

**1** **Variations in the chemical composition of the submicron aerosol and in the**

**2** **sources of the organic fraction at a regional background site of the Po Valley**

**3** **(Italy)**

**5** **M. Bressi[1], F. Cavalli[1], C. A. Belis[1], J. -P. Putaud[1], R. Fröhlich[2], S. Martins dos Santos[1], E. Petralia[3], A. S.**

**6** **H. Prévôt[2], M. Berico[3], A. Malaguti[3] and F. Canonaco[2]**

**8** [1]European Commission, Joint Research Centre, Institute for Environment and Sustainability, Air and

**9** Climate Unit, Via Enrico Fermi 2749, Ispra (VA) 21027, Italy.

**10** [2]Paul Scherrer Institute, Laboratory of Atmospheric Chemistry, Villigen 5232, Switzerland.

**11** [3]Italian National Agency for New Technologies, Energy and Sustainable Economic Development (ENEA),

**12** Via Martiri di Monte Sole 4, Bologna 40129, Italy.

**14** Correspondence to:

**15** michael.bressi@jrc.ec.europa.eu, claudio.belis@jrc.ec.europa.eu, fabrizia.cavalli@jrc.ec.europa.eu

**17** **Abstract**

**18** Fine particulate matter (PM) levels and resulting impacts on human health are in the Po Valley (Italy)

**19** among the highest in Europe. To build effective PM abatement strategies, it is necessary to characterize

**20** fine PM chemical composition, sources and atmospheric processes on long time scales (>months), with

**21** short time resolution (<day), and with particular emphasis on the predominant organic fraction.

**22** Although previous studies have been conducted in this region, none of them addressed all these aspects

**23** together. For the first time in the Po Valley, we investigate the chemical composition of non-refractory

**24** submicron PM (NR-PM$_1$) with a time-resolution of 30 minutes at the regional background site of Ispra

**25** during one full year, using an Aerosol Chemical Speciation Monitor (ACSM) under the most up-to-date

**26** and stringent quality assurance protocol. The identification of the main components of the organic

**27** fraction is made using the Multilinear-Engine 2 algorithm implemented within the latest version of the

**28** SoFi toolkit. In addition, with a view of a potential implementation of ACSM measurements in European

**29** air quality networks as a replacement of traditional filter-based techniques, parallel multiple off-line

**30** analyses were carried out to assess the performance of the ACSM in the determination of PM chemical

**31** species regulated by Air Quality Directives. The annual NR-PM$_1$ level monitored at the study site (14.2



µg/m$^3$) is among the highest in Europe, and is even comparable to levels reported in urban areas like
New York City (USA, 14.2 µg/m$^3$) and Tokyo (Japan, 12-15 µg/m$^3$). On the annual basis, submicron
particles are primarily composed of organic aerosol (OA, 58% of NR-PM$_1$). This fraction was apportioned
into oxygenated OA (OOA, 66%), hydrocarbon-like OA (HOA, 11% of OA), and biomass burning OA
(BBOA, 23%). Among the primary sources of OA, biomass burning (23%) is thus bigger than fossil fuel
combustion (11%). Significant contributions of aged secondary organic aerosol (OOA) are observed
throughout the year. The unexpectedly high degree of oxygenation estimated during wintertime is
probably due to the contribution of secondary BBOA and the enhancement of aqueous phase
production of OOA during cold months. BBOA and nitrate are the only components of which
contributions increase with the NR-PM$_1$ levels. Therefore, biomass burning and NO$_x$ emission reductions
would be particularly efficient in limiting submicron aerosol pollution events. Abatement strategies
conducted during cold seasons appear to be more efficient than annual-based policies. In a broader
context, further studies using high-time resolution analytical techniques on a long-term basis for the
characterization of fine aerosol should help better shape our future air quality policies, which constantly
need refinement.

1.  Introduction
The Po Valley region - located in northern Italy - is amongst the most polluted areas in Europe (van
Donkelaar et al., 2010; EEA, 2013). Annual PM$_{2.5}$ (particulate matter with an aerodynamic diameter
below 2.5 µm) mean concentrations can significantly exceed the European PM$_{2.5}$ annual limit value (25
µg/m$^3$ in 2015, European Directive 2008/50/EC) and the recommendations of the World Health
Organization (PM$_{2.5}$ annual average of 10 µg/m$^3$; WHO, 2006) at urban (e.g. Bologna, 35.8 µg/m$^3$) and
regional background sites (e.g. Ispra, 32.2 µg/m$^3$; Putaud et al., 2010). Consequently, PM$_{2.5}$ impacts on
human health are among the most severe in Europe (EC, 2005), while impacts on the local radiative
forcing are substantial (Clerici and Mélin, 2008; Ferrero et al., 2014; Putaud et al., 2014b). Effective PM
abatement strategies are thus needed in the Po Valley and require an in-depth knowledge of the
chemical composition of fine PM, to quantify its sources and the atmospheric processes leading to its
secondary formation.

In this region, high levels of fine aerosol are mostly due to the conjunction of i) high pollutant

emissions related to industrial, transport, biomass burning and agricultural activities - the Po river basin
hosting 37% of the Italian industries, 55% of the livestock and contributing 35% of the Italian agricultural
production (WMO et al., 2012) - and ii) the specific geography and topography of this area - a flat basin





surrounded by the Alps and Apennine Mountains dominated by weak winds that favour the accumulation of pollutants (Decesari et al., 2014; Kukkonen et al., 2005; Pernigotti et al., 2012). As a consequence, PM levels are not only high in urban areas but also at regional and rural background sites, which are key locations for investigating air pollution due to their distance from local sources and local phenomena. Measurements of fine PM mass and chemical composition at rural background sites are in addition specifically required in the current European Directive on air quality (2008/50/EC).

Previous studies have investigated the properties of fine aerosols at regional and rural background sites of the Po valley region, including their chemical characteristics (e.g. Carbone et al., 2014; Putaud et al., 2002, 2010; Saarikoski et al., 2012), and their main sources (Belis et al., 2013; Gilardoni et al., 2011; Larsen et al., 2012; Perrone et al., 2012). Fine aerosols are primarily made of organics (30-80% of fine PM mass, depending on the site and season studied), followed by ammonium nitrate and ammonium sulfate. Their main sources are fossil fuel, biomass burning and biogenic emissions to name a few. In studies dealing with long time-series (entire season or year), the chemical composition of fine aerosol is generally measured with a relatively low time resolution (typically 24 hours), thus preventing from studying its diurnal variation and short-lived chemical-physical processes. When documented with higher time-resolutions (1 hour or less), aerosol chemistry is usually determined for intensive campaigns of a few weeks only, hence not suitable to depict the seasonal or yearly air quality situation. In addition, the complexity of the fine organic fraction (e.g. Jimenez et al., 2009) requires state-of-the-art analytical and source apportionment (SA) techniques to identify organic aerosol chemical properties and sources.

The recently developed Aerosol Chemical Speciation Monitor (ACSM, Aerodyne Research Inc., Ng et al., 2011a) is suitable to fill these gaps by providing the chemical composition of non-refractory submicron aerosols (NR-PM$_1$) with a time resolution of 30 min, while operating on long time scales. Even though promising results have been recently reported (e.g. Budisulistiorini et al., 2014; Canonaco et al., 2013, 2015; Minguillón et al., 2015; Ng et al., 2011a; Petit et al., 2015; Ripoll et al., 2015; Sun et al., 2012), this technique remains recent and requires additional field deployment to test its consistency with independent methods for the monitoring of fine PM chemistry (e.g. filter measurements). In addition, information on the accuracy of this technique is of paramount importance given the growing number of ACSMs in Europe and the necessity to build a network of quality assured and harmonized instruments for comparability of results – at present about 20 ACSMs are in operation in Europe (http://www.psi.ch/acsm-stations/overview-full-period) within the frame of the EU ACTRIS network (Aerosols, Clouds, and Traces gases Research InfraStructure, http://www.actris.eu/). Moreover, by using



receptor models, the apportionment of organic aerosol (OA) into its major components - hydrocarbon-
like (HOA), biomass burning (BBOA) and oxygenated OA (OOA) - can be performed (Lanz et al., 2007;
Zhang et al., 2011 and references therein).

In this study, we used an ACSM during one year with a 30 min time-resolution at a regional

background site of the Po Valley and performed subsequent SA analyses with the aim of: i) describing
the high time resolved chemical composition of NR-PM$_1$ on a long time-scale, to better understand the
physicochemical processes driving its temporal variations, ii) apportioning the organic fraction into its
main sources, iii) identifying PM abatement strategies to efficiently reduce NR-PM$_1$ pollution events at
regional background areas of the Po valley, and iv) assessing the atmospheric consistency of ACSM
measurements when compared to independent analytical methods, to evaluate its possible
implementation in future European Air Quality networks.

2.   Material and methods

2.1. Sampling site

Measurements were conducted at the European Commission – Joint Research Centre (EC-JRC) Ispra site
(45°48'N, 8°38'E, 217 m a.s.l.), which is part of the European Monitoring and Evaluation Programme
(EMEP) measurement network (http://www.nilu.no/projects/ccc/ sitedescriptions/it/index.html) and
the        Global        Atmosphere        Watch        (GAW)        regional        stations
(http://www.wmo.int/pages/prog/arep/gaw/measurements.html). It is located on the northwest edge
of the Po Valley region, 60 km northwest of the Milan urban area. It can be regarded as a "regional/rural
background" site following the criteria recommended by the European Environment Agency (Larssen et
al., 1999). For simplicity, the term "regional background site" will be used in the following although
comparisons with rural background sites from other studies will also be reported. Further information
on the study site can be found in Putaud et al. (2014b).

2.2. Aerosol Chemical Speciation Monitor (ACSM)

The recently developed ACSM (Aerodyne Research Inc., ARI) was used to measure the non-refractory
(NR) chemical composition (organics, nitrate, sulfate, ammonium, chloride) of submicron particles (PM$_1$)
with a 30 minutes time resolution. The operating principle of the ACSM is similar to the widespread
Aerodyne aerosol mass spectrometer (Canagaratna et al., 2007; Jayne et al., 2000), with the difference
that the former does not inform on the size distribution of the chemical composition of NR-PM$_1$. A full
description of the ACSM can be found in Ng et al. (2011a). Briefly, an aerodynamic lens is used to focus



submicron particles (50% transmission range of 75-650 nm; Liu et al., 2007), which are then vaporized in
high vacuum, ionized by electron ionization (at 70eV) and detected by a quadrupole mass spectrometer
(Pfeiffer Vacuum Prisma Plus RGA). Two different quadrupole-ACSMs (Q-ACSMs) were used in this study
(from March 2013 to February 2014): Q-ACSM#1 from 01 March to 18 August 2013 and Q-ACSM#2 from
20 June 2013 to 28 February 2014. Note that Q-ACSM#2 was not running from 3 November to 18
December due to its participation in the first inter-ACSM comparison exercise (Crenn et al., 2015). The
reproducibility and consistency with independent measurements are discussed in Sect. 3.1. In the
following, orthogonal regressions are reported unless otherwise stated.

Both ACSMs were operated with the latest Data Acquisition (DAQ 1.4.3.8 to 1.4.4.5) and Data

Analysis (DAS 1.5.3.0 to 1.5.3.2) software (ARI, https://sites.google.com/site/ariacsm/mytemplate-sw)
available at the time of use, which are developed within Igor Pro 6.32A (Wavemetrics).
Recommendations given by Aerodyne (2010a, 2010b) and Ng et al. (2011a) were followed for the
operation, calibration and data analysis of the ACSMs. Ammonium nitrate calibrations were performed
seasonally and used for the determination of experimental nitrate response factors (RF) and ammonium
relative ionization efficiencies (RIE, see Sect. S1 for further details). Annual average and season-
dependent experimental RF and RIE values were alternatively applied to assess whether the ACSM is
stable over multi-seasonal periods (see Sect. 3.1 for results). RIEs for organics, nitrate and chloride (1.4,
1.1 and 1.3, respectively) were taken from the literature (Canagaratna et al., 2007; Takegawa et al.,
2005). RIE for sulfate was experimentally determined based on ammonium sulfate calibrations for
ACSM#2, and was taken from the literature for ACSM#1 (see Sect. S1). Collection efficiencies (CE) set as
i) a fixed 0.5 value (e.g. Budisulistiorini et al., 2013) or ii) following the composition-dependent CE
algorithm introduced by Middlebrook et al. (2012) were compared in order to determine the most
appropriate CEs (see Sect. 3.1 for results).

2.3. Additional analytical techniques

Additional measurements routinely performed at the JRC-Ispra site are used in this study (see Putaud et
al., 2014a for a full description). $PM_{2.5}$ are sampled on quartz fibre filters (Pall, 2500 QAT-UP) with a
Partisol PLUS 2025 sampler equipped with a carbon honeycomb denuder operating at 16.7 L/min from
01 March 2013 to 28 February 2014 with daily filter changes at 08:00 UTC. Major ions ($NH_4^+$, $K^+$, $NO_3^-$,
$SO_4^{2-}$, etc.) are analysed by ion chromatography (Dionex DX 120 with electrochemical eluent
suppression) after extraction in Milli-Q water (Millipore). Organic and elemental carbon (OC and EC,
respectively) are quantified by a thermal-optical method (Sunset Dual-optical Lab Thermal-Optical



Carbon Aerosol Analyzer) using the EUSAAR-2 protocol (Cavalli et al., 2010). Equivalent Black Carbon
(BC) is measured by a Multi Angle Absorption Photometer (MAAP, Thermo Scientific, model 5012)
applying an absorption cross section of 6.6 m$^2$/g of equivalent black carbon at the operation wavelength
of 670 nm. Particle volume concentrations are determined with a home-made Differential Mobility
Particle Sizer (DMPS) combining a Vienna-type Differential Mobility Analyser (DMA) and a Condensation
Particle Counter (CPC, TSI 3010), following the European Supersites for Atmospheric Aerosol Research
(EUSAAR) specifications for DMPS systems (Wiedensohler et al., 2012). Meteorological variables
(temperature, pressure, relative humidity, precipitation, wind speed and direction) are determined from
a weather transmitter WXT510 (Vaisala, Finland). Solar radiation is measured by a CM11 pyranometer
(Kipp and Zonen, The Netherlands).

2.4. Apportionment of the organic fraction

The organic fraction was apportioned using the Positive Matrix Factorization approach (PMF, Lanz et al.,

2007; Paatero and Tapper, 1994; Ulbrich et al., 2009; Zhang et al., 2011) by applying the Multilinear
Engine 2 algorithm (ME-2, Paatero, 2000) implemented in the SoFi tool (v4.8, Canonaco et al., 2013;
Crippa et al., 2014). Details on the theory and application of PMF and ME-2 can be found in the
aforementioned studies. Briefly PMF aims at factorizing an initial X matrix (representing the temporal
variation of *m/z* signals here) into two F and G matrices (representing factor profiles and contributions,
respectively) putting a constraint of non-negativity on F and G matrices. Contrary to classical program
used to resolve PMF, ME-2 allows any element of the F and G matrices to be constrained with a certain
degree of freedom. This ME-2 approach has been typically used to constrain full factor profiles (e.g.
Amato et al., 2009; Crippa et al., 2014), specific elemental ratios (e.g. Sturtz et al., 2014) or specific
species contribution (e.g. Crawford et al., 2005) in a given factor profile.

In our study, ME-2 is applied with and without constraining factor profiles (FPs), using the so-

called *a*-value approach (Canonaco et al., 2013) in the former case, which can be described as follows:
$(f_{k,j})_{solution}=(f_{k,j})_{reference} \pm a.(f_{k,j})_{reference}$                                      (1)
where k and j are the indexes for the factors and the species, respectively, $f_{k,j}$ is the element (k, j) of the
F matrix, the index "solution" stands for the PMF user solution, "reference" for the reference profile and
"*a*" is a scalar defined between 0 and 1 (e.g. applying an *a*-value of 0.10 lets ±10% variability to our FP
solution with respect to the reference FP). Following Crippa et al. (2014), we perform a sequence of runs
with i) unconstrained PMF, ii) fixed HOA, iii) fixed HOA and BBOA, iv) fixed HOA, BBOA and cooking OA
(COA) factors before selecting the most appropriate solution. Uncertainties are calculated using the DAS



1.5.3.0 version following the methodologies of Allan et al., 2003a and Ulbrich et al. (2009). *m/z* 12 and
13 are removed for SA analysis since negative signals are observed most of the time. Reference factor
profiles (RFPs) are taken from ambient deconvolved spectra from the Aerosol Mass Spectrometry (AMS)
spectral database (Ulbrich et al., 2015). HOA and BBOA profiles are taken from Ng et al. (2011c) (average
of profiles from multiple studies) and COA from Crippa et al. (2013). Different *a*-values are tested (see
Sect. 3.2) applying i) relative standard deviations of averaged RFPs defined for every *m/z* (i.e. assuming
that the chosen averaged RFPs are representative of our data set), ii) recommendations of Crippa et al.
(2014) based on the SA of 25 European AMS data sets and iii) comparison with independent
measurements (e.g. NOx, CO, BC, etc.). Solutions from 2 to 8 factors are investigated in order to choose
the appropriate number of factors (see Sect. S2 and 3.2).

3.  Results

3.1. Consistency of ACSM measurements

Ammonium nitrate calibrations performed on each ACSM are shown in
Figure S1. $RF_{NO3}$ and $RIE_{NH4}$ do not present significant seasonal variability - e.g. for ACSM#2 $RF_{NO3}$=4.7E-
11±0.2E-11 A.$\mu g^{-1}$.$m^3$ - , suggesting constant calibration factors may be used throughout the campaign.
On the other hand, calibration factors exhibit substantial discrepancies between both ACSMs (e.g. $RF_{NO3}$
of 2.5E-11 and 4.7E-11 A.$\mu g^{-1}$.$m^3$ for ACSM#1 and #2, respectively), suggesting that instrument-specific
factors are necessary. Applying constant and composition-dependent CEs does not lead to noticeable
differences (e.g. for NR-$PM_1$: $r^2$=0.97, slope=1.00±0.00, y-intercept=0.10±0.03 $\mu g/m^3$, n=14842) due to i)
low sampling line RH (e.g. typically below 30% for ACSM#2), and ii) few high-nitrate-content events (only
5% of data exhibits ammonium nitrate mass fractions>40%, defined as high by Middlebrook et al., 2012).
The Middlebrook et al. (2012) algorithm is however preferred since slightly acidic aerosols are observed
at the study site (on average sulfate plus nitrate against ammonium in $\mu eq/m^3$: $r^2$=0.96,
slope=1.21±0.00, intercept=0.01±0.00 $\mu eq/m^3$, n=14842).

A comparison performed between the two ACSMs used in this study during a 2-month summer

period is shown in Figure S2. Very good correlations are observed for every chemical component
(0.91<$r^2$<0.98, n=1402, hourly average) - chloride excluded - with slopes relatively close to one
(0.87<slopes<1.42), indicating a fairly good comparability between both instruments. One of the two
ACSM also participated in the first-ever inter-ACSM comparison exercise performed between 13
different European Q-ACSMs during 3 weeks in Paris, France (Crenn et al., 2015). Satisfactory
performances - defined by |z-scores|<2 - are reported for our instrument regarding every chemical





component and NR-PM$_1$ mass, attesting the consistency of our measurements with other European
sites.
Measurements performed by the ACSM and independent off-line and on-line analytical
techniques are compared in Figure 1 and Table 1. An overall good agreement is found for every major
components throughout the year (typically $r^2$>0.8), although discrepancies are observable for specific
species and seasons. On the annual scale, a good agreement ($r^2$=0.77, n=317) is found between organics
from ACSM and OC from filter measurements in spite of expected filter sampling artefacts (Maimone et
al., 2011; Turpin et al., 2000; Watson et al., 2009). Even better agreements are observed on a seasonal
basis ($r^2$~0.9), with steeper slopes in summer compared with winter, which likely reflects the different
degrees of oxygenation of organics among seasons (leading to different OM-to-OC ratios). However,
these slopes cannot be directly regarded as the OM-to-OC ratios due to i) differences in size fractions
between both methods (PM$_1$ for ACSM and PM$_{2.5}$ for filter measurements) and ii) uncertainties related
to RIE$_{Org}$ for ACSM measurements (Budisulistiorini et al., 2014; Ripoll et al., 2015). An estimation of the
OM-to-OC ratio for submicron organics applying the methodology described by Canagaratna et al.
(2015) is discussed in Sect. 4.2. Good correlations are observed for nitrate during all seasons ($r^2$>0.9) but
summer ($r^2$=0.5), which is most likely related to enhanced evaporative losses of ammonium nitrate from
filter during the latter season (Chow et al., 2005; Schaap et al., 2004). Slopes range from 0.9 to 1.4 -
summer excluded - which is comparable to what is reported elsewhere (Budisulistiorini et al., 2014;
Crenn et al., 2015; Ripoll et al., 2015). Very good correlations are observed for sulfate in every season
($r^2$=0.9-1.0) with slopes close to 1 (0.9-1.1, winter excluded), consistent with its presence in the
submicronic size fraction and its low volatility leading to the minimization of sampling artefacts. Note
that discrepancies have been reported when comparing sulfate measured by the ACSM (Petit et al.,
2015) or the AMS (Zhang, 2005) with independent measurements. Our results suggest that ammonium
sulfate calibrations should be performed to experimentally determine sulfate RIEs, which appear to be
instrument-specific but stable over several months. Although aerosols are slightly acidic on average at
the study site, ammonium mostly neutralizes nitrate and sulfate throughout the campaign and thus
exhibits behaviour in between the two latter compounds. Higher uncertainties are associated with
chloride from filter quantification, resulting in no agreement with ACSM measurements in summer
when the concentrations are the lowest ($r^2$=0.00), and fairly good agreement during the other seasons
($r^2$=0.64-0.77). The high slope observed for the ACSM#1 (e.g. 2.1 during spring) compared to the fairly
good slopes observed for ACSM#2 (0.7-1.1) suggests that chloride RIE might be instrument-specific and



require appropriate calibrations for its accurate quantification (see also Riffault et al., 2013 on this
topic).
The sum of NR-PM$_1$ components and BC has been compared to the volume concentration of
PM$_1$. Good agreement is found between both variables ($r^2$>0.8) giving further confidence on the
consistency of our ACSM measurements. The annual average particle density estimated from this
comparison (i.e. slope) is 1.6, which is typical of ambient aerosol particles densities (1.5-1.9 in Hand and
Kreidenweis, 2002; Hu et al., 2012; McMurry et al., 2002; Pitz et al., 2003, 2008). The higher densities
observed during spring and summer (1.9-2.0) than autumn and winter (1.3-1.5) are likely due to the
enhanced contribution of secondary aerosol and aged particles during the former period (Pitz et al.,

2008).


3.2. Organic apportionment quality control
First, during the aforementioned inter-ACSM comparison study (Crenn et al., 2015), source
apportionment of organics was performed based on data from 13 Q-ACSMs (Fröhlich et al., 2015),
including one ACSM used in the present study. Satisfactory performances (|z-scores|<2) are reported for
our ACSM using a similar approach as adopted in this study. This result demonstrates that our
instrument and the associated data treatment, including the source apportionment modelling, are
capable of accurately identifying and quantifying OA sources.

3.2.1. Model configurations
Regarding our specific study, the configuration applied to reach the optimal SA of organics is
thoroughly discussed in Sect. S2 (constrained factor profiles, number of factors, $a$-values and
integration-period durations). Briefly, constraining both HOA and BBOA factors result in satisfactory
solutions with relevant factor profiles, time series and daily cycles. Other configurations (e.g.
unconstrained factors) lead to unsatisfactory results with high seed variability, mixing of factors or
absence of key fragments in identified profiles (e.g. absence of $m/z$ 43 and 44 in BBOA contrary to what
is reported in Heringa et al., 2011, Figure S3). Solutions applying different number of factors are
investigated. Three-factors (HOA, BBOA and OOA) are retained during spring, autumn and winter
whereas two factors (HOA and OOA) are most suitable during summer. A lower number of factors
results in a mixing of them, whereas a higher number generates additional factors - e.g. semi-volatile
OOA (SV-OOA) during summer, OOA-BBOA during autumn - which are not satisfactory - e.g. missing
fragments or poor correlations with external data, see Table S1. BBOA cannot be clearly identified





during summer i.e. in this season agricultural waste burning contributions are estimated to be minor
(maximum 3-4% of OA, Sect. S2). Note that COA could not be evidenced, likely due to the type of site
studied (regional background) and the lower sensitivity, time- and mass-to-charge-resolution of the
ACSM compared to classical AMS instruments (further discussed in Sect. S2). Uncertainties associated
with factor contributions are estimated by performing sensitivity tests on $a$-values, which are regarded
as the most subjective input parameters. Five scenarios putting very low to very high constraints on the
reference factor profiles have been defined (see Table S2). Comparable solutions in terms of relative
contributions (Figure S4) and agreement with independent measurements (Table S2) are found when
applying low to high constraints following the empiric recommendations of Crippa et al. (2014).
Unsatisfactory solutions are generally reached under the extreme scenarios (fully fixed factor profiles
and $m/z$ specific standard deviations of reference factor profiles). We decided to apply low constraints
(i.e. $a$-values of 0.1 and 0.5 for HOA and BBOA, respectively) to let as much freedom as possible to our
factor profiles while remaining in the range of plausible solutions. SA was performed on 3-months, 6-
months and 1-year datasets. Although comparable solutions are found for each configuration (number
of factors, factor profiles, diurnal cycles, comparisons with external data), applying SA on seasonal
datasets was preferred since i) the seasonal variability of factor profiles is captured and ii) questionable
results are observed in summer for 6-months and 1-year configurations (see Sect. S2). When comparing
the sum of OA factor concentrations and measured OA on the annual scale, OA is very well modelled
($r^2$=0.97, slope=0.98±0.00, intercept=0.1±0.0 µg/m$^3$, n=14842).

3.2.2.   Model optimal solution

Factor profiles, contributions and daily cycles of the optimal SA solution are presented in Figure

2. Independent factor profiles and time series are found for each season, which is a prerequisite for
having reliable SA solutions. HOA is identified during every season and exhibits a profile dominated by
alkyl fragments such as $m/z$ 55 (from the $C_nH_{2n-1}^+$ ion series) and $m/z$ 57 (from $C_nH_{2n+1}^+$ ion series; Ng et
al., 2011c). Its relative contribution peaks in the morning and is higher during weekdays than weekends,
which is characteristic of traffic emissions. BBOA is found during every season but summer and has a
profile similar to that of HOA, except for the high contribution of $m/z$ 60 ($C_2H_4O_2^+$) and 73 ($C_3H_5O_2^+$),
which have been suggested as biomass burning markers (Lee et al., 2010 and references therein). A
distinct daily cycle with higher contributions during night-time than daytime is observed, in addition to
higher contributions during weekends than weekdays, consistent with residential heating emissions. The
low BBOA concentrations modelled during late spring and early autumn, as well as the small increased



contribution observed during the morning also suggest residential heating emissions. OOA is identified
thanks to the predominant contribution of $m/z$ 44 ($CO_2^+$) and 43 ($C_2H_3O^+$). The higher contribution of $f_{44}$
(defined as $m/z$ 44 to total organic signal; 0.17-0.23 depending on seasons) with respect to $f_{43}$ (defined
similarly; 0.05-0.09) suggests that this OOA factor is highly oxidized and presents low volatility (LV-)
rather than semi-volatility (SV-) OOA characteristics (see Jimenez et al., 2009 and Zhang et al., 2011 for
definitions of these components). This statement is supported by very good correlations ($r^2$=0.96-0.99)
found between our unconstrained OOA profiles and the average low-volatility OOA (LV-OOA) profile
reported by Ng et al. (2011c) from 6 AMS studies. Interestingly, our OOA profiles present slight seasonal
differences that likely reflect changes in source contributions and/or physical-chemical processes in this
factor. For instance, $f_{60}$ in OOA profiles is enhanced in winter (0.014) compared with other seasons
(0.001-0.004), which suggests that biomass burning contributes to this factor during the aforementioned
season, consistent with different studies reporting $f_{60}$ in secondary OA from biomass burning (e.g.
Cubison et al., 2011; Heringa et al., 2011; see Sect. S2). Daily cycles are comparable for all seasons with a
bimodal pattern characterized by a small peak during night-time and a prominent peak during daytime.
The latter peak suggests that a fraction of (LV-) OOA could be locally rather than regionally produced on
the time scale of few hours only, likely due to enhanced photochemical activities during daytime. The
former peak could be due to i) the condensation of highly oxygenated semi-volatile material favoured by
night-time thermodynamic conditions or ii) a contribution of SV-OOA in our OOA factor, which is
generally dominated by LV-OOA. The absence of an $f_{44}$ night-time peak (Sect. 4.2) suggests that the
second assumption is most probable implying that both SV-OOA and LV-OOA influence our OOA factor.

3.2.3.   Time series comparisons

Comparisons between our OA factors, $m/z$ tracer and independent species time series are

shown in Table 2. OOA time series show very good agreement with Org_43 (organic signal at $m/z$ 43)
and Org_44 ($r^2$>0.8 and 0.9, respectively) and relatively good agreement with secondary inorganic
species (e.g. $r^2$≥0.5 for ammonium), indicating that this factor can be regarded as a surrogate for
secondary organic aerosols. Comparisons with sulfate (a low-volatility species) and nitrate (a semi-
volatile species) confirm that our OOA factor might be a mix of SV- and LV-OOA, since better agreement
is found with one or the other compound depending on the season studied. BBOA exhibits very good
coefficients of determination when compared with its presumable fragment tracers Org_60 and Org_73
($r^2$>0.97), giving further confidence on its appropriate quantification. Good correlations are generally
found between BBOA and BC ($r^2$≥0.5, except for summer) indicating that a large proportion of BC stems





from biomass burning, consistent with previous findings at the study site (Gilardoni et al., 2011, from EC
measurements). A good agreement is also observed with CO ($r^2 \geq 0.7$), as already reported in the Alpine
valleys (e.g. Gaeggeler et al., 2008). HOA is not as well correlated with external data or specific $m/z$,
which could be related to i) the absence of clear $m/z$ tracers for this factor due to similarities with BBOA
profile, ii) the absence of clear external tracers due to co-emissions by fossil fuel and biomass burning
activities of BC, CO and NOx and iii) possible uncertainties associated with the apportionment between
HOA and BBOA. The first two assumptions are attested by the better agreement observed between HOA
and $m/z$ fragments or independent data during summer (e.g. $r^2=0.52$, n=2208 between HOA and BC) and
specific months (e.g. May, September), when biomass burning contributions are negligible. Although
uncertainties associated with the accurate apportionment of HOA and BBOA cannot be excluded,
several pieces of evidence indicate that a mixing of both factors is unlikely since HOA and BBOA present
i) independent factor time series during all seasons ($r^2=0.1-0.2$), ii) distinct and relevant daily cycles and
iii) no significant $a$-value variability attesting their robustness.

4.   Discussion

The meteorological representativeness of this one-year measurement is assessed by comparing the
solar irradiation, precipitation, and temperature monthly averages to the ones measured during 1990-
2010 at the study site (Figure S5). Comparable seasonal averages are generally found in our study and
during the bidecadal reference period. Nevertheless compared to 1990-2010, Spring 2013 was rainier,
Summer 2013 slightly warmer and sunnier, and Winter 2013-2014 rainier. Further information regarding
the representativeness of measurements performed at the study site during the year 2013 can be found
in Putaud et al. (2014a).

4.1. Chemical composition of NR-PM$_1$

An overview of the chemical composition of NR-PM$_1$ retrieved during this campaign is shown in Figure 3.
The annual averaged NR-PM$_1$ mass reported here (14.2 µg/m$^3$) ranges amongst the highest NR-PM$_1$
levels reported at rural and urban downwind sites in Europe (Crippa et al., 2014) and worldwide
(Jimenez et al., 2009; Zhang et al., 2007, 2011). It is comparable to NR-PM$_1$ levels reported during
specific campaigns in the urban areas of New York City (USA, 12 µg/m$^3$, Weimer et al., 2006), Tokyo
(Japan, 12-15 µg/m$^3$, Takegawa et al., 2006) or Manchester (UK, 14 µg/m$^3$, Allan et al., 2003a, 2003b).
Our annual average NR-PM$_1$ mass is higher than the 10 µg/m$^3$ guidelines given by the World Health
Organization for the annual average PM$_{2.5}$ mass (including refractory and non-refractory compounds;





WHO, 2006). After similar conclusions have been drawn for $PM_{2.5}$ and $PM_{10}$ size fractions (Putaud et al.,
2010), the Po Valley appears to be one of the most polluted regions in Europe with regard to $NR\text{-}PM_1$
levels this time. Submicron aerosol particles are mostly made of organics (58%), nitrate (21%), sulfate
(12%) and ammonium (8%). The predominance of organics is typical of urban downwind sites (e.g.
average of 52% reported in Zhang et al., 2011). On the other hand, the noticeable proportion of nitrate
is characteristic of urban sites (18% in Zhang et al., 2011), which likely reflects the substantial influence
of anthropogenic activities emissions at our regional site. As a result, sulfate exhibits particularly low
contributions at the study site compared with other locations (generally >20% in Zhang et al., 2011).
NR-PM$_1$ levels present a clear seasonality with higher levels during spring (~18 µg/m$^3$) and
winter (~15 µg/m$^3$) compared with summer and autumn (~12 µg/m$^3$). Higher levels were expected
during cold months due to enhanced biomass burning emissions, lower boundary layer heights (BLH)
and previous observations (Putaud et al., 2013). Expected seasonal variations of the chemical
composition of NR-PM$_1$ are observed, with i) higher nitrate contributions during the cold season which
favours its partitioning in the condensed phase (Clegg et al., 1998), ii) higher sulfate contributions during
summer associated with enhanced photochemical production (Seinfeld and Pandis, 2006), and iii)
relatively stable contributions for ammonium (mainly neutralizing the two previous species) and
organics (discussed later on).
A focus will now be made on daily cycles of the chemical composition of NR-PM$_1$ (Figure 4),
displayed for the first time during the 4 seasons in the Po Valley, thanks to the high time resolution and
stability of the ACSM. On the annual scale, daily cycles of NR-PM$_1$ levels are characterized by
significantly higher concentrations during night time than daytime, likely due to lower BLH, higher wood
burning emissions (during cold seasons) and lower temperatures favouring the partitioning of semi-
volatile inorganic (mainly ammonium nitrate) and organic material in the condensed phase, to name a
few. A distinct peak is however observed around noon, probably caused by enhanced photochemical
production of secondary inorganic (mostly ammonium sulfate) and organic compounds, especially
during summer (Figure S6). Note that this annual daily pattern is the combination of distinct daily cycles
varying with the season studied (Figure S6). In terms of relative chemical composition, organics are
dominating NR-PM$_1$ mass independently of the time of the day, with median contributions ranging from
~60 to 70%. Nitrate exhibits higher contribution during night time due to its abovementioned semi-
volatile nature. Sulfate shows unexpected daily cycles with significantly different (99.99% confidence
level) relative contributions - and absolute concentrations - during daytime (~15% of NR-PM$_1$ mass
around noon) compared to night time (~10% around midnight, Figure 4), although its formation was





expected to occur mainly over longer time scales (i.e. days) in cloud droplets (Ervens et al., 2011). This
observation could be due to i) local production of sulfate with increased photochemical production
around noon at the study site and/or ii) diurnal changes of the atmospheric stratification in the Po Valley
as described by Saarikoski et al. (2012), enhancing aged particle contribution during the middle of the
day and the afternoon. Non-refractory chloride (mostly $NH_4Cl$, Huang et al., 2010) exhibits very low
contributions independently of the hour of the day (medians below 0.5% of $NR-PM_1$ mass) with however
a slight increase at night, which is likely due to its presumable semi-volatile nature here. Note that 30
min averaged chemical composition is also presented in Figure 3, capturing sudden variations of $PM_1$
levels and chemical composition throughout the year, highlighting the velocity at which changes in
chemical processes and source emissions occur.

4.2. Focus on organic aerosols
An overview of the contribution of HOA, BBOA and OOA to OA is shown in Figure 5. On the annual
average, the organic fraction is dominated by the secondary component (OOA, 66%). Although this OOA
contribution is substantial, higher proportions are generally reported at rural and urban downwind sites
worldwide (90 and 82% of OA on average, respectively, Zhang et al., 2011). This lower relative
contribution of OOA is related to the higher contribution of (primary) BBOA in our study (23% of OA on
the annual average) compared to the previous ones. Considerable contributions of BBOA are explained
by the specific location of the study site in the vicinity of the Alps, where biomass burning is a major
contributor to OA (Belis et al., 2011; Herich et al., 2014; Lanz et al., 2010). Biomass burning emissions
hence substantially affect OA levels on the annual scale here. The contribution of HOA is comparatively
smaller (11%), indicating that despite the expected large contributions of fossil fuel emissions (i.e. traffic
and industrial emissions), those are not the major sources of primary OA at the study site. On the other
hand, it is likely that fossil fuel emissions of volatile organic compounds (VOCs) - which are OOA
precursors - contribute to our OOA levels as reported elsewhere (Gentner et al., 2012; Volkamer et al.,
2006). At the study site, Gilardoni et al. (2011) previously estimated on the basis of [14]C analyses that
secondary organic carbon stemming from fossil emissions might represent 12% of OC on the annual
average. In other words, fossil fuel emissions could represent approximately a quarter (12+11=23%) of
total OA mass when both primary and secondary OA fractions are accounted for. The analysis of the
components' seasonal variations show relatively stable HOA contributions (9-14%), higher contributions
of BBOA during cold seasons due to residential heating (up to 36% of OA on average during winter) and





higher OOA contributions during summer related to enhanced photochemical production (86% of OA on
average).
OA can be further characterized investigating specific organic fragments. $m/z$ 44 (mainly $CO_2^+$)
and 43 (mainly $C_2H_3O^+$) signals give insights on the nature of OA, as the former is primarily related to
acids or acid-derived species whereas the latter is mostly associated with non-acid oxygenates (Duplissy
et al., 2011; Ng et al., 2011b). Daily variations of both $f_{44}$ and $f_{43}$ are shown in Figure 6, along with other
major organic fragments. On average, $f_{44}$ is predominant with respect to $f_{43}$ (15 and 7%, respectively),
which indicates that acid species dominate the OA composition with respect to non-acid oxygenates.
Both fragments present different daily patterns underlying distinct mechanisms of formation. Acids'
contributions are enhanced during daytime, which could be explained by photochemical processes
and/or daily BLH variations as already discussed for sulfate. Non-acid oxygenates exhibit higher
contributions during night time than daytime, although the amplitude of their daily cycles is less
pronounced than that of acid species (~1 and 3%, respectively). This pattern could be due to i) the
formation of semi-volatile non-acids during night time by e.g. condensation favoured by low
temperatures as previously found for semi-volatile OOA (Lanz et al., 2007), ii) their degradation during
daytime by e.g. fragmentation reactions (Daumit et al., 2013) and/or iii) their conversion into acid-
related species during daytime by e.g. functionalization or oligomerization reactions (Daumit et al.,
2013). It should be specified that the enhancement of $f_{44}$ during daytime (~2%) and the decreasing of $f_{43}$
during night-time (~1%) only represent a small fraction of their total contributions to OA (medians of 13-
16% and 7-8% depending on the time of the day, respectively, Figure 6), suggesting that most acid and
non-acid oxygenates have been formed before reaching our sampling site, i.e. have been imported from
other regions. The other major OA fragments ($m/z$ 29, 55, 57 and 60) present i) constant contributions
for $f_{29}$ due its various emission sources (HOA, BBOA, OOA; Ng et al., 2011c), ii) the absence of lunch peak
for $f_{55}$ (and also for the absolute contributions of $m/z$ 55) consistent with the presumable low influence
of cooking emissions, iii) morning and evening peaks for $f_{57}$ characteristics of fossil fuel emissions and iv)
higher contributions during night time for $f_{60}$ in agreement with its biomass burning origin.
Using $f_{43}$ and $f_{44}$, the oxygen-to-carbon (O/C), OM-to-OC (OM/OC), hydrogen-to-carbon (H/C)
ratios and the carbon oxidation state (OSc) have been estimated for total OA based on the
methodologies described by Aiken et al. (2008), Kroll et al. (2011) and Ng et al. (2011b), and applying the
parameterization defined in Canagaratna et al. (2015), which can be summarized as follows:
O/C= 4.31 $f_{44}$+ 0.079                                                  (1)
OM/OC= 1.28 O/C+ 1.17                                            (2)





$H/C = 1.12 + 6.74\,f_{43} - 17.77\,f_{43}^{2}$                                                                          (3)
$OSc = 2 * O/C - H/C$                                                                                                        (4)
with H/C (and therefore OSc) being estimated only if $f_{44} > 0.05$ and $f_{43} > 0.04$ (Canagaratna et al., 2015).
The errors (average absolute value of the relative error) are estimated to be 28%, 8% and 13% for O/C,
OM/OC and H/C for standard molecules, respectively, and 0.06 units for OSc (Canagaratna et al., 2015).
Discrepancies in $f_{44}$ quantifications between different ACSMs, and between ACSMs and AMS (Crenn et
al., 2015; Fröhlich et al., 2015) are however likely to increase the uncertainties associated with O/C,
OM/OC and OSc estimates. In particular, in the abovementioned inter-instrument comparison, ACSMs
have been reported to overestimate $f_{44}$ by up to a factor of 2 compared to the HR-ToF-AMS (Fröhlich et
al., 2015). Comparisons with studies using (HR-ToF-) AMS instruments will thus not be reported and only
variations within this dataset will be discussed. Regarding the two ACSMs used in this study, absolute
differences between median estimates from two-months co-located measurements in summer are 0.05,
0.07, 0.07 and 0.17 for O/C, OM/OC, H/C and OSc, respectively (see Figure S7). Seasonal and annual O/C,
OM/OC, H/C and OSc are shown in Figure 7. High O/C, OM/OC and OSc are found on the annual scale
(medians of 0.7, 2.1 and -0.2, respectively), reflecting once more the aged, oxidized properties of
organic matter at the study site, consistent with the predominance of the OOA component. Little
seasonal variations are observed for the aforementioned variables hence highlighting the high degree of
oxidation of OA throughout the year (Figure 7). The unexpectedly high degree of oxygenation of OA
observed during cold seasons despite the increased contribution of primary BBOA (with OM/OC ratios of
1.4-1.6) could be explained by the contribution of secondary BBOA in our OOA factor during these cold
seasons, which could be associated with the enhancement of e.g. dicarboxylic and ketocarboxylic acid
contents (Kundu et al., 2010) that have extremely high OM/OC ratios (up to 3.8 and 3.1, respectively,
Turpin and Lim, 2001). This assumption is supported by the higher proportion of $f_{60}$ in our OOA factor
(discussed in Sect. 3.2 and S2), as well as the surprisingly high OM/OC ratio observed for OOA during
winter (2.5 compared to 2.2-2.4 during the other seasons). Note that Canonaco et al. (2015) also report
a higher $f_{44}$ in (LV-) OOA in winter compared to summer in Zurich (Switzerland). According to these
authors, this could be due to enhanced aqueous-phase production of (LV-) OOA in clouds or hygroscopic
aerosols in winter, which would lead to higher levels of oxygenation compared to gas-phase oxidation
mechanisms typically occurring during summer.

4.3. Possible implications for PM abatement strategies





In order to investigate the characteristics of fine aerosol pollution events, the variations of NR-PM$_1$
chemical composition and OA factors' contributions as a function of total NR-PM$_1$ mass are examined
(Figure 8). Distinct trends are observed depending on the chemical species and OA components studied.
The proportion of nitrate is clearly enhanced with increasing NR-PM$_1$ levels (from ~10 to >30% when
[NR-PM$_1$]>30 μg/m$^3$) indicating that nitrate - or NO$_x$ - abatement policies should be highly effective when
attempting to limit PM$_1$ pollution events in the Upper Po Valley. Sulfate shows an opposite trend with
decreasing relative contribution when NR-PM$_1$ mass increases (e.g. <5% when [NR-PM$_1$]> 50 μg/m$^3$),
likely due to the lower concentrations of sulfate during cold seasons, when the highest number of
pollution events is observed. The proportion of organics is substantial (48-66%) independently of NR-
PM$_1$ mass, justifying once again the importance of determining its sources to design adequate
abatement policies. When focusing on the organic fraction, BBOA is the only OA factor exhibiting
increased contributions (from ~10 to >40%) with increased NR-PM$_1$ mass (from <10 to >60 μg/m$^3$),
which points out the PM abatement potential of effective biomass burning emission reductions. HOA
levels are rather constant throughout the year and therefore their proportions steadily decrease when
NR-PM$_1$ levels increase, implying that local fossil fuel related emissions of primary OC are not the main
responsible for submicron pollution events observed at the study site. Although OOA always represents
a major fraction of OA (41-75% depending on the mass bin studied), its contribution steadily decreases
with increasing NR-PM$_1$ mass. This unexpected result signifies that even though aged, secondary,
oxidised organics are the main contributor to OA on the annual average (66%), they do not play a
prominent role in fine PM acute pollution events.

Current European legislations set daily and/or annual PM limit values depending on the size fraction

addressed (Directive 2008/50/EC). Volume size distributions suggest that approximately 90% of the
PM$_{2.5}$ mass concentration is borne by particles below an aerodynamic diameter of 1 μm at the study site
(Putaud et al., 2014a). Therefore, measures tackling the main constituents of the submicron aerosol
fraction would be efficient for complying with PM$_{2.5}$ legislations. Based on the chemical characterization
of NR-PM$_1$ and SA of its organic fraction with a time-resolution of 30 min over 1-year, this study provides
new evidence which could orient PM abatement strategies also at similar regional background sites of
the Po Valley. On the annual scale, OA and especially OOA should be of main concern given their
predominance in NR-PM$_1$ chemical composition (Figure 3). On the seasonal scale, efforts should be
directed towards the cold seasons (winter and early spring), for which the highest NR-PM$_1$ levels are
observed, due to specific meteorological conditions (e.g. low BLH, low temperatures) and emission
sources (e.g. biomass burning, Figure 3 and Figure 5). In particular, measures addressing emissions of



NO$_x$ and BBOA would be the most efficient for reducing the magnitude and frequency of PM pollution
events (Figure 8).

Recommendations for PM abatement strategies are formulated here from a legislative perspective,

which aims at decreasing PM levels. Although diminishing PM levels should help reducing PM impacts,
the existence of a direct causal relationship can be debatable since each chemical component has a
specific effect on human health (WHO, 2013), the radiative forcing (Boucher et al., 2013) or ecosystems
(e.g. Carslaw et al., 2010). For instance, implementing policies aiming at mitigating nitrate
concentrations - as suggested previously in this section - would likely have limited health benefits
according to toxicological studies (Reiss et al., 2007; Schlesinger and Cassee, 2003), and should lead to
an increased global warming (Boucher et al., 2013). On the other hand, measures reducing BBOA levels
should be beneficial, since the cardio-vascular effects of biomass burning particles have been widely
reported in the literature (Bølling et al., 2009; Miljevic et al., 2010; Naeher et al., 2007) and could be
similar to those of traffic-emitted particles (WHO, 2013 and references therein), whereas their impacts
on the radiative forcing could be null (Boucher et al., 2013). Strategies aiming at reducing solely PM
mass are therefore limited, and an assessment of their impacts - e.g. using integrated assessment
models (Carnevale et al., 2012; Janssen et al., 2009) with appropriate parameterizations of fundamental
processes - would be beneficial.

5.   Conclusion and perspectives

The NR-PM$_1$ chemical composition and the apportionment of the organic fraction have been
investigated for the first time with this completeness at a regional background site of the Po valley
(Italy), using high time-resolution (30 min) and long term (1 year) measurements with a state-of-the-art
quality assured ACSM and the most advanced factor analysis methods. Comparisons between two
ACSMs show very good time series correlations for the major compounds (0.91<$r^2$<0.98, n=1402) with
however discrepancies in their absolute concentrations (0.9<slopes<1.4). These results are promising
with regard to the consistency of ACSM measurements at different locations, but also underlines the
importance of conducting inter-ACSM comparisons to define common protocols and assure data
comparability among the European ACSM network (see Crenn et al., 2015). Comparisons between ACSM
and independent analytical technique measurements show an overall good agreement for major
components throughout the year (typically $r^2$>0.8). Discrepancies observed in time series correlations
and quantifications (i.e. slopes) for specific species and seasons (e.g. nitrate in summer) are attributed
to filter sampling artefacts. These results are encouraging regarding the potential implementation of



ACSMs in air quality networks as a replacement of traditional filter-based techniques, to measure the
artefact-free chemical composition of fine aerosols with high time-resolution. Additional comparison
studies are nevertheless needed to support our results, and further technical development allowing the
refractory carbon fraction to be accounted for is required.
NR-PM$_1$ and PM$_1$ levels measured in the upper Po Valley (14.2 and 15.3 µg/m$^3$ on the annual
average, respectively) are among the highest reported in Europe, stressing the need for implementing
effective PM abatement strategies in this region. On average, the chemical composition of non-
refractory submicron aerosol is dominated by organic aerosol (58% of NR-PM$_1$), which is composed of
HOA (11% of OA), BBOA (23%) and OOA (66%). Fossil fuel combustion is thus not a major source of
primary OA in this area of the Po Valley. Primary BBOA significantly contributes to OA on the annual
average and especially during winter (36%). Our OOA component is highly oxidised and aged with an LV-
OOA spectral signature, a large proportion of acid-related species and high OM/OC ratios. Highly
oxidised OA properties are observed during all seasons, surprisingly including winter, which could reflect
secondary BBOA influence and OOA aqueous-phase formation processes during cold seasons. Further
research aiming at identifying the sources of OOA - including secondary BBOA using e.g. high resolution
mass spectrometric techniques (Crippa et al., 2013) or proton nuclear magnetic resonance (Paglione et
al., 2014) - and better estimating O/C, OM/OC and OSc parameters would be beneficial.
Specific recommendations for PM abatement strategies at a regional level can be suggested.
The higher frequency of particulate pollution peaks observed during cold seasons suggests an
orientation of future policies towards these periods. BBOA and nitrate present increasing relative
contributions with increasing fine aerosol levels, which suggests that wood burning and NO$_x$ emission
reductions should notably decrease NR-PM$_1$ pollution events. Note that these recommendations are
only formulated in the perspective of reducing PM levels, assuming a subsequent reduction of PM
impacts. Additional dimensions - e.g. specific impacts of each chemical component, short versus long-
term exposure, co-benefit of sanitary and climatic impacts - should also be considered when defining
PM abatement strategies. In a broader context, the use of high time resolution analytical techniques for
the measurement of PM pollution properties can help better shape our future air quality policies.

*Acknowledgements.* This study was partially supported by the European Union's project ACTRIS
(Aerosols, Clouds, and Trace gases Research InfraStructure Network, EU FP7-262254). R. Passarella (EC-
JRC), K. Douglas (EC-JRC), V. Pedroni (EC-JRC) and M. Stracquadanio (ENEA) are thanked for their help on
the field and/or for the chemical analyses of filters. P. Croteau (Aerodyne) is acknowledged for his



technical support on the operation of the ACSM. N. Jensen (EC-JRC) is thanked for providing gas phase
data. M. Crippa (EC-JRC) is acknowledged for her valuable advices.



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





**Tables and Figures**

Table 1. Consistency of ACSM measurements: comparison between ACSM and independent analytical techniques using orthogonal regression
analyses. Slopes and intercepts are indicated ± uncertainties.

| | $r^2$ | | | | | slope | | | | | intercept | | | | |
|---|---|---|---|---|---|---|---|---|---|---|---|---|---|---|---|
| | Sp | Su | Au | Wi | An | Sp | Su | Au | Wi | An | Sp | Su | Au | Wi | An |
| Org vs OC | 0.91 | 0.90 | 0.86 | 0.92 | 0.77 | 2.18 ± 0.07 | 2.92 ± 0.10 | 1.87 ± 0.09 | 1.26 ± 0.04 | 1.72 ± 0.04 | -0.29 ± 0.37 | -1.07 ± 0.32 | -0.28 ± 0.36 | 0.74 ± 0.37 | 0.61 ± 0.25 |
| Nitrate | 0.95 | 0.53 | 0.96 | 0.92 | 0.91 | 1.37 ± 0.03 | 4.27 ± 0.25 | 1.28 ± 0.03 | 0.86 ± 0.03 | 1.28 ± 0.02 | 0.42 ± 0.18 | 0.64 ± 0.11 | 0.48 ± 0.10 | 0.62 ± 0.11 | 0.48 ± 0.09 |
| Sulfate | 0.96 | 0.97 | 0.92 | 0.86 | 0.95 | 1.05 ± 0.02 | 0.98 ± 0.02 | 0.96 ± 0.04 | 1.38 ± 0.06 | 1.00 ± 0.01 | -0.01 ± 0.04 | 0.02 ± 0.06 | 0.04 ± 0.07 | -0.25 ± 0.06 | 0.00 ± 0.03 |
| Ammonium | 0.92 | 0.70 | 0.91 | 0.95 | 0.90 | 1.03 ± 0.03 | 1.00 ± 0.06 | 0.93 ± 0.04 | 0.81 ± 0.02 | 0.99 ± 0.02 | -0.04 ± 0.07 | -0.04 ± 0.07 | -0.12 ± 0.05 | 0.03 ± 0.03 | -0.08 ± 0.03 |
| Chloride | 0.75 | 0.00 | 0.59 | 0.78 | 0.52 | 2.68 ± 0.13 | -0.13 ± 0.09 | 0.68 ± 0.06 | 1.13 ± 0.07 | 1.75 ± 0.06 | 0.04 ± 0.01 | 0.03 ± 0.00 | 0.04 ± 0.00 | -0.02 ± 0.01 | 0.02 ± 0.01 |
| Mass vs volume | 0.87 | 0.82 | 0.88 | 0.85 | 0.81 | 1.91 ± 0.01 | 1.95 ± 0.02 | 1.45 ± 0.01 | 1.34 ± 0.01 | 1.63 ± 0.01 | -1.16 ± 0.19 | -1.36 ± 0.18 | -2.45 ± 0.19 | -0.11 ± 0.20 | -1.09 ± 0.11 |



Legend: Sp: spring (March-April-May), Su: summer (June-July-August), Au: autumn (September-October-November), Wi: winter (December-
January-February), An: annual. Independent analytical techniques refer to i) EC-OC Sunset Analyzer for OC from $PM_{2.5}$ sampling, ii) Ion
Chromatography for ions from $PM_{2.5}$ sampling and iii) DMPS for volume concentrations (see Sect. 2.3 for more details). Mass refers to NR-
$PM_1$+BC. Intercepts are in µg/m$^3$. Slopes of mass vs volume are in g/cm$^3$ and dimensionless otherwise.



Table 2. Comparison (coefficient of determination, $r^2$) between SA factors, organic $m/z$ tracers and independent species time series. BC stands
for Black Carbon; Org_i stands for organic signal at $m/z$ i (i=43, 44, 60, 67, 73, 81).

| | | HOA | | | | | BBOA | | | | | OOA | | | |
|---|---|---|---|---|---|---|---|---|---|---|---|---|---|---|---|
| | Org_67 | Org_81 | NOx | CO | BC | Org_60 | Org_73 | NOx | CO | BC | Org_43 | Org_44 | NH4 | SO4 | NO3 |
| SPRING | 0.60 | 0.55 | 0.03 | 0.08 | 0.28 | 0.99 | 0.97 | 0.32 | 0.81 | 0.70 | 0.88 | 0.94 | 0.76 | 0.43 | 0.77 |
| SUMMER | 0.90 | 0.91 | 0.07 | 0.40 | 0.52 | | | - | | | 0.97 | 0.94 | 0.54 | 0.60 | 0.19 |
| AUTUMN | 0.63 | 0.61 | 0.07 | 0.10 | 0.24 | 0.99 | 0.97 | 0.06 | 0.68 | 0.47 | 0.82 | 0.92 | 0.47 | 0.53 | 0.38 |
| WINTER | 0.58 | 0.57 | 0.34 | 0.33 | 0.39 | 0.98 | 0.97 | 0.20 | 0.66 | 0.63 | 0.80 | 0.99 | 0.50 | 0.39 | 0.66 |






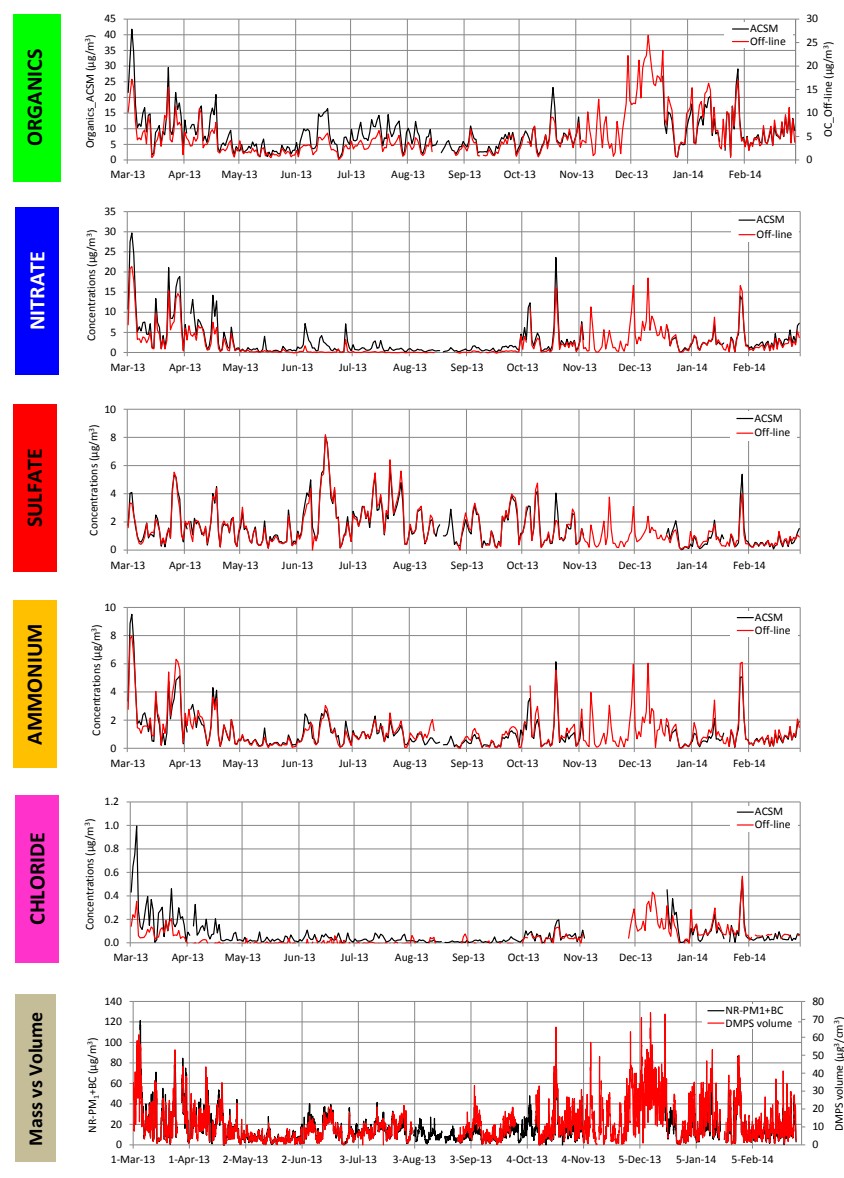


Figure 1. Comparison between measurements performed with the ACSM and other co-located analytical techniques. See Table 1 and Sect. 2.3 for more details.






Figure 2. Organic source apportionment presented by season: factor profiles (left), time series (middle)
and daily cycles (right, error bars represent 1 standard deviation). Seasons are defined as Spring: MAM,
Summer: JJA, Autumn: SON and Winter: DJF.



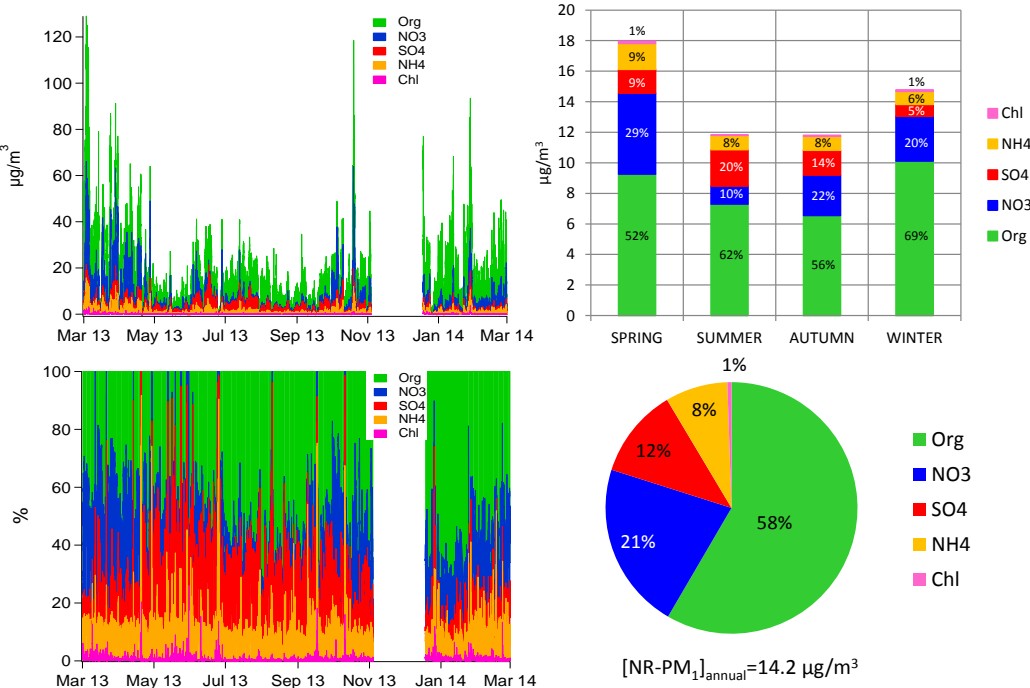


Figure 3. Overview of the chemical composition of NR-PM$_1$ at Ispra (Po Valley, Italy). Left: absolute (top)
and relative (bottom) chemical composition with 30 min time resolution; top right: absolute seasonal
average, bottom right: annual average.



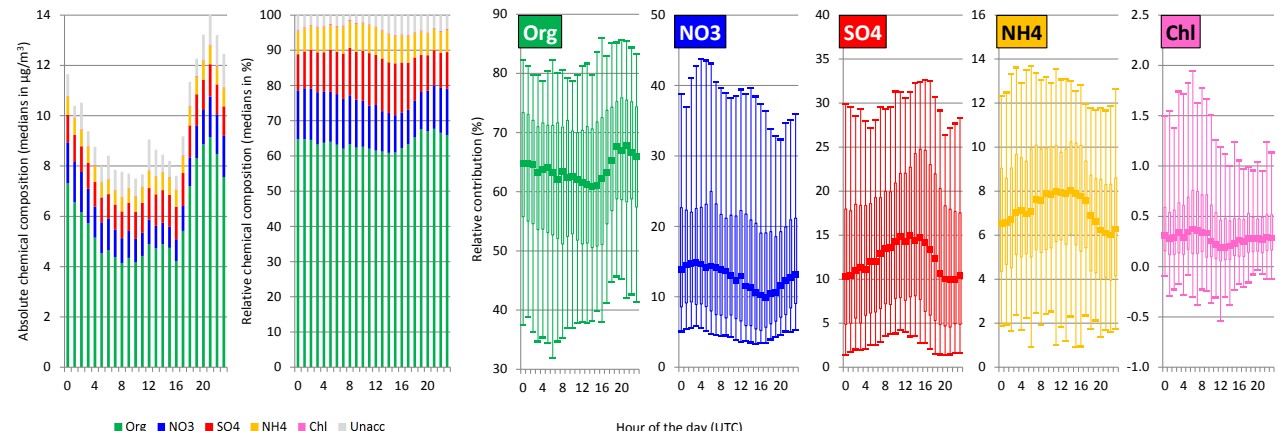


Figure 4. Daily cycles of NR-PM$_1$ chemical composition on the annual scale. Unacc: unaccounted mass, whisker plots are constructed from the
5[th], 25[th], 50[th], 75[th] and 95[th] percentiles.





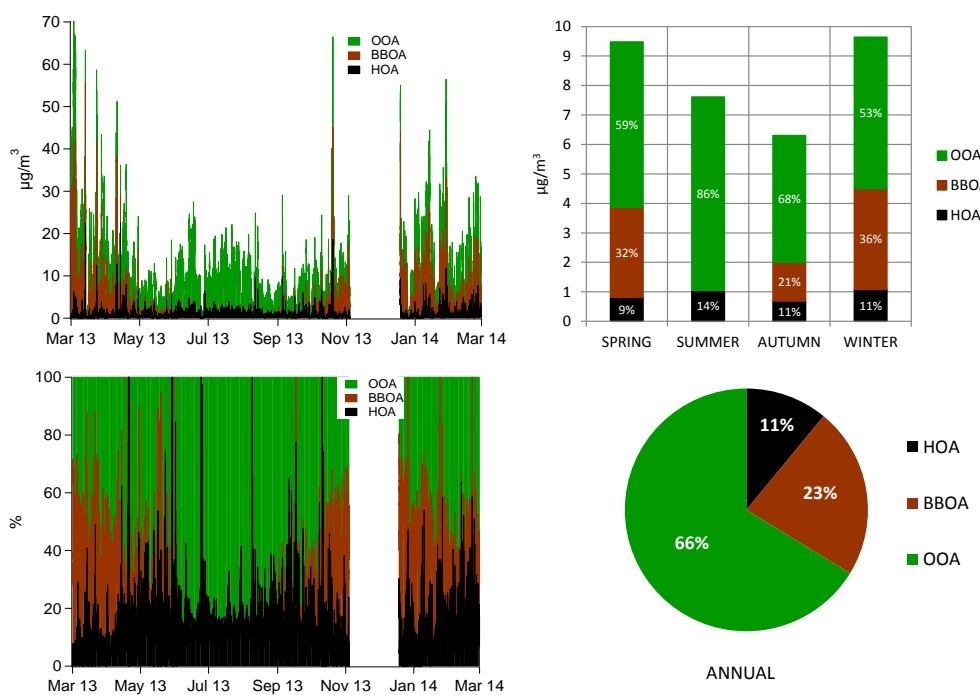


Figure 5. Overview of HOA, BBOA and OOA contributions to organic aerosols; see legend Figure 3.



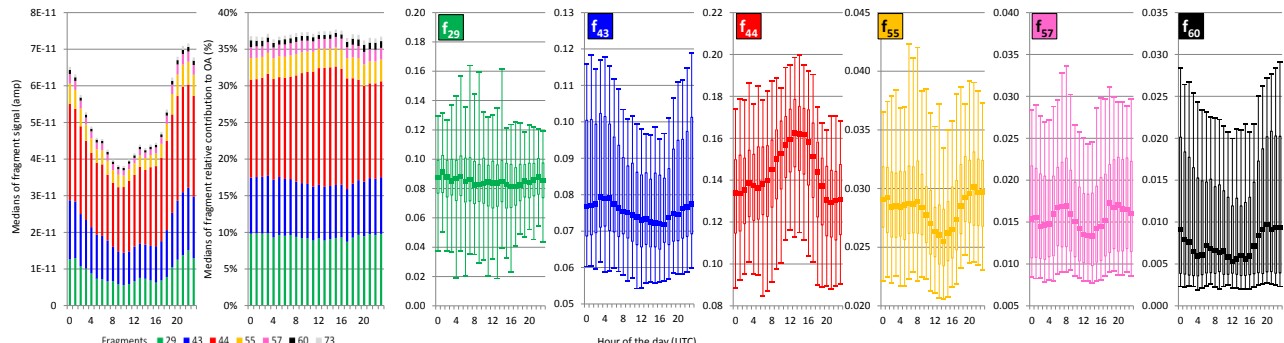


Figure 6. Annual statistics describing the daily cycles of the major organic fragments. Box plots are constructed from the 5[th], 25[th], 50[th], 75[th] and

95[th] percentiles.





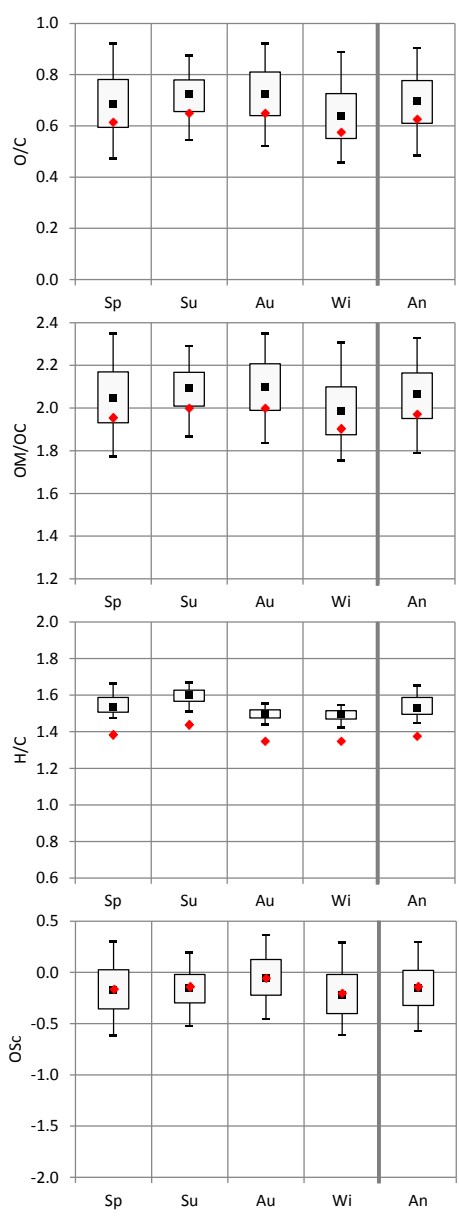


Figure 7. Seasonal and annual O/C, OM/OC, H/C and OSc of ambient OA. Sp: spring (MAM), Su: summer
(JJA), Au: autumn (SON), Wi: winter (DJF), An: annual. Black: 5[th], 25[th], 50[th], 75[th] and 95[th] percentiles
estimates following Canagaratna et al. (2015); red: median estimates following Aiken et al. (2008) for
O/C and OM/C, Ng et al. (2011b) for H/C and Aiken et al. (2008), Kroll et al. (2011) and Ng et al. (2011b)
for OSc. Note that the authors do not recommend comparing absolute O/C, OM/OC and OSc values
reported here with other AMS studies, given the uncertainties associated with $f_{44}$ quantifications from
ACSM measurements (please see text).

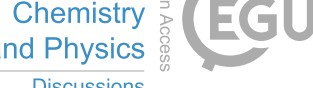

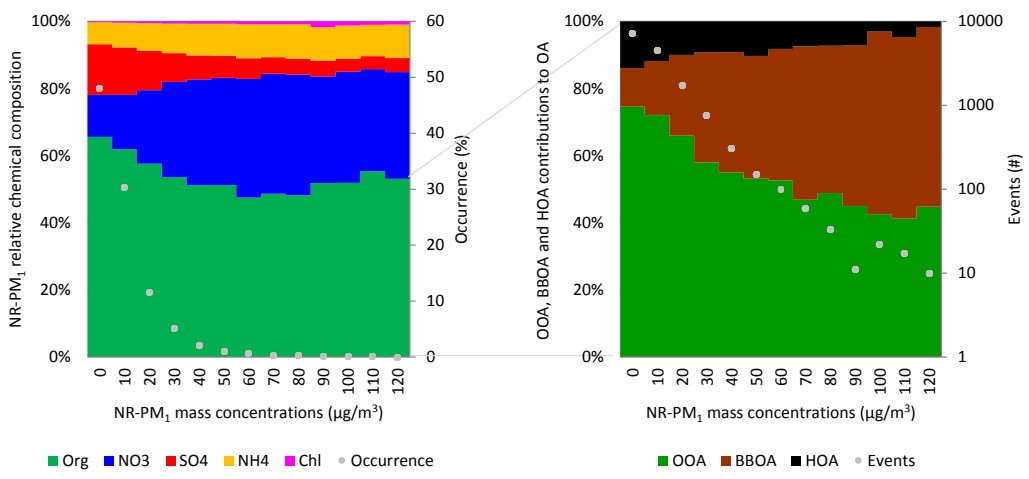


Figure 8. NR-PM$_1$ relative chemical composition (left) and OA factor contributions (right) averages in function of NR-PM$_1$ mass concentrations (bins of 10 μg/m$^3$). Occurrence (%, left) and number of events (#, right) are indicated (solid dots) for each NR-PM$_1$ bin. Note that one event corresponds to one 30 minute average.