# Peer review of "Variations in the chemical composition of the submicron aerosol and in the 1 sources of the organic fraction at a regional background site of the Po Valley 2 (Italy) 3 4 M. Bressi1, F. Cavalli1, C. A. Belis1, J. -P. Putaud1, R. Fröhlich2, S. Mar"

_Atmospheric Chemistry and Physics, 2016_

## Referee Comment (RC1) · Anonymous Referee #1 · 13 Mar 2016

The manuscript by Bressi et al. analyzed one-year submicron aerosol data that was collected at a regional background site in Po Valley using an ACSM. The performance of the ACSM measurements was fully evaluated, and the variations in chemical composition and diurnal cycles were characterized. Further, positive matrix factorization with ME-2 algorithm was used to resolve potential organic aerosol factors with different sources and processes. While SOA was important throughout the year, an enhanced role of biomass burning aerosols during cold seasons was also observed. Atmospheric aerosols in Po Valley have been widely characterized at both urban and rural sites and the results in this study were overall consistent with previous conclusions. Although

there are no new scientific findings compared to previous studies, this manuscript is still worth for publication by providing a general overview in aerosol variations and sources on an annual basis in Po Valley.

My major comment is the discussions on seasonal differences. In the text, the figures only showed the averages on annual basis (some seasonal averages in supplementary). I suggest the authors move the important seasonal information into the main text. This is also one of the uniqueness of this study. For example, show the diurnal cycles of NR-PM1 species during four seasons in Figure 4. Please also show the diurnal cycles of mass concentrations of OA factors in Figure 2, sometimes, it is misleading if only the diurnal cycles of mass fractions are shown.

Fog events are frequent in Po Valley. I am surprised that the authors didn't discuss the fog impacts (scavenging and production) on aerosol variations, particularly in winter.

With one-year data, weekend effects might be explored for a better interpretation of sources.

Line 394, "and previous observations (Putaud et al., 2013)" is not one of the reasons. Line 449, typo, "C2H30+".

---

## Referee Comment (RC2) · Anonymous Referee #2 · 16 Mar 2016

**Referee comment on**

**"*Variations in the chemical composition of the submicron aerosol and in the sources of the organic fraction at a regional background site of the Po Valley (Italy)*"**

**by M. Bressi et al.**

This manuscript reports results obtained with an Aerodyne Aerosol Chemical Speciation Monitor (ACSM) during a long-term measurement period (1 year) in the Po Valley (Italy). The authors investigated the chemical composition of non-refractory submicron PM (NR-PM$_1$) with a time-resolution of 30 minutes and identified of the main components of the organic fraction. In addition, the parallel multiple off-line analyses were carried out to assess the performance of the ACSM in the determination of PM chemical species regulated by Air Quality Directives. This work is meaningful, and the workload is very large. The observed results are representative because they have been measured for almost one year. It is completely within the scope of Atmospheric Chemistry and Physics. However, the results presented here are a little superficial, and a more deeply analyses on these observation results is necessary. Therefore, I recommend its publication after a major revision.

Specific Comments

Abstract

Although the abstract has summarized the main contents of the text, it is still lack of some points to attract readers. I think the authors have done a lot of works, but they do not have an in-depth analysis on their results. For example, the author did not find the BBOA in summer. While, they only showed the average results in this part and main text. Then some interesting information was averaged and masked. Therefore, I believe other more meaningful results could be found after a more in-depth analysis.

Introduction

Please add a simple introduction on the previous ACSM and AMS research results in this area. Then discuss the innovation of your research is more appropriate.

About the article structure.

Section 3 (results) contains : (1) discussion on consistency of ACSM measurements; (2) discussion on organic apportionment quality control. Meanwhile, some content looks like a introduction of analysis method. I think these contents are more like a discussion part not results. On the contrary, the author showed a lot of experimental results in section 4 (discussion). Therefore, I suggest that the author reconsider the article structure carefully and some discussion content could be put in the supplementary material.

About the language: Some sentences are not easy to understand. While high quality language expression is very important for a high quality article. Therefore, please improve

the language after this modification.

At the beginning of section 4.3, the variations of NR-PM$_1$ chemical composition and OA factors' contributions as a function of total NR-PM$_1$ mass are examined. However, these analysis is not deep enough. According to the results mentioned above, the contribution of every NR-PM$_1$ and OA species to the total is different in four seasons. It means that the pollution characteristics in four seasons may be different. Therefore, I strongly recommend the author to discuss the variation of every species with the change of total NR-PM$_1$ mass for different seasons. Then some common features for four seasons and some unique characteristics for one season could be found. These results are more interesting than that provided now.

Figure 3:
(1) "Left, bottom, top right, bottom right" is not clear enough. Figure 3a, b, c, d is better. I suggest the author do the modification in this and other figures.
(2) This figure looks not every clear.
For example:
The following figure is a part of the relative chemical composition of all NR-PM1 species. I found that the contribution of organic in spring (March-April-May) is lower than 50% (red line) in most of time, and frequently lower than 40%. However, the average contribution reached to 52% in the main text. It looks very strange.
In addition, according to the relative chemical composition figure, it seems that the contribution of organic was 0 some time. However, the mass concentration of organic looks always higher than 0.

[Figure]

The same phenomenon also appeared in Figure 5. Please see the following figure:

[Figure]

It can be found that the contribution of HOA is almost always higher than 10% (red line), even frequently higher than 30%. However, the average contribution was only 11%.

Other comments:

Abstract.

Line 31-33. A detail comparison with other study results is not necessary in abstract. I don't recommend to cite the detailed data results.

Introduction.

Line 50 Please provide the annual $PM_{2.5}$ mean concentration value. That could make this sentence more convincing.

Line 69 "2008/50/EC" is a reference or other standard ? It is not easy to understand.

Section 2.1 I suggest the author add a sampling site map in main text or supplement material.

Line 205 Please check the format.

Line 219 What is the reason for the poor correlation and results difference of chloride ?

Section 3.2.3 Although the author explained the reasons for HOA was not well correlated with BC, CO and NOx, I still feel confused. What are the main sources of BC, CO, and NOx in the study site ? It looks BBOA has a very big impact on them. In addition, chloride is also an important tracer for BBOA, did you have an analysis of their relationship ?

Line 376-378 Pleases add the values of the highest NR-$PM_1$ levels reported at rural and urban .......

Line 385-386 Please add "(Fig. 3)" to the back of this sentence.

Line 393 Did you measured the BLH in different seasons ? A comparison on measurement results is more useful.

Figure 1 and 2. These figures are very difficult to see clearly due to the too small words. It is necessary to enlarge the font size in the two figures.

Figure 6. It is not very clear and the font is too small. I suggest that all the figures are divided into two lines. Then the figures will not too narrow.

---

## Referee Comment (RC3) · Anonymous Referee #3 · 21 Mar 2016

This paper presents an analysis of ACSM data from the Po Valley, Italy, using ME-2 factorisation. While this is becoming an increasingly common form of measurement and analysis, this paper remains relevant because it is the first such analysis of one of the most polluted rural sites in Europe and may have implications for regional pollution in this area. The results aren't particularly surprising and the technical developments are incremental at best, however the results and quality assurance data are presented in a very comprehensive manner and the results analysed in the context of air quality control policy, so it makes an overall contribution to the science in these regards. Overall, this paper is well-written and I would recommend publication after the authors

consider the following minor comments.

General comments:

The paper currently lacks a comparison with other AMS factorisations done at this site, specifically Decesari et al. (2014, doi:10.5194/acp-14-12109-2014) and Dall'Osto et al. (2015, doi:10.1021/acs.est.5b02922). This strikes me as a major omission.

I am struggling to see what the mass spectral marker analysis in section 4.2 contributes to the conclusions of the paper. The analysis exhibits behaviours broadly similar with the results from the factorisation and while speculative conclusions are offered for the behaviours, these are largely inconclusive. The section could do with being shorter and more focused on the analyses that result in new scientific insight.

Specific comments:

Line 328: Given that m/z=60 results from a primary emissions from biomass burning, it is possible that its presence in OOA is more likely due to factor mixing than SOA production. Such an issue is very possible, given the variations within BBOA and the mass spectral resemblance of primary HULIS to LV-OOA (e.g. http://www.atmos-chem-phys.net/15/2429/2015/).

Line 349: While a good correlation between the BBOA factor with 60 and 73 is worth reporting, this only indicates that the factor follows these markers; to take this as a sign of accuracy, one must assume that these markers are accurate reflections of actual BBOA, which may or may not be the case.

Line 362: None of these tests do not exclude the possibility that there is a degree of rotational freedom between factors. Such freedoms can change magnitudes of signals without significantly changing their time series. It's also possible that exchanges between factors can be via a third factor (e.g. OOA).

Line 397: Other reasons for a seasonal high of sulphate are plausible, such as changes in source regions due to seasonal changes in the prevailing wind direction, or changes

in the amount of rainout.

Line 407: The statement that the midday peak is due to in situ photochemistry is at odds with the discussion towards the end of the paragraph, where this is rightfully treated with scepticism. This could be tested by comparing SO4 with SO2 and looking for a diurnal pattern in the fraction of oxidised sulphur as SO4. However, I would expect it more likely that the peak is due to the increased PBL height during the day favouring downward mixing of advected pollution. As pointed out, the timescale of formation of SO4 is too long to expect a pattern like this to result from chemistry.

Technical comments:

While conventional, the definitions of the seasons used should be given in the main text rather than just a figure caption.

Line 178: Please specify the 'classical program' used for PMF.

SI line 20: Please use scientific rather than engineering notation.

SI line 102: Correct "Error! Reference source not found."

Figure S5: Do the curved lines on these plots represent actual data or a nonlinear interpolation between points? If it is the latter, the algorithm should be specified and justified.
* * *

---

## Author Comment (AC1) · 29 Jul 2016

Dear Referee #1,

Thank you for your comments on our manuscript. Please find below your comments in blue and the authors' responses in black.

The manuscript by Bressi et al. analyzed one-year submicron aerosol data that was collected at a regional background site in Po Valley using an ACSM. The performance of the ACSM measurements was fully evaluated, and the variations in chemical composition and diurnal cycles were characterized. Further, positive matrix factorization with ME-2 algorithm was used to resolve potential organic aerosol factors with different sources and processes. While SOA was important throughout the year, an enhanced role of biomass burning aerosols during cold seasons was also observed. Atmospheric aerosols in Po Valley have been widely characterized at both urban and rural sites and the results in this study were overall consistent with previous conclusions. Although there are no new scientific findings compared to previous studies, this manuscript is still worth for publication by providing a general overview in aerosol variations and sources on an annual basis in Po Valley.

My major comment is the discussions on seasonal differences. In the text, the figures only showed the averages on annual basis (some seasonal averages in supplementary). I suggest the authors move the important seasonal information into the main text. This is also one of the uniqueness of this study. For example, show the diurnal cycles of NR-PM1 species during four seasons in Figure 4.

The important seasonal information has been moved into the main text accordingly. Figure 4 has therefore been modified and now show the diurnal cycles of NR-PM$_1$ species during four seasons and on the annual scale. Figure S6 has been removed. Following referee #2 suggestions as well, the variations of NR-PM$_1$ chemical composition and OA factors' contributions as a function of total NR-PM$_1$ mass are also examined on the seasonal scales (Fig. S9 and Sect. S4).

Please also show the diurnal cycles of mass concentrations of OA factors in Figure 2, sometimes, it is misleading if only the diurnal cycles of mass fractions are shown.

Figure 2 has been modified accordingly and now show the diurnal cycles of mass concentrations of OA factors.

Fog events are frequent in Po Valley. I am surprised that the authors didn't discuss the fog impacts (scavenging and production) on aerosol variations, particularly in winter.

Fog events are indeed frequent in the Po Valley. We decided not to discuss this subject in the manuscript for two main reasons. First, our manuscript is focused on statistically significant variations (e.g. seasonal, daily) in order to suggest recommendations for PM abatement strategies. We thus decided not to comment specific pollution events (e.g. biomass burning, traffic or fog events). Second, previous studies have reported in detail the fog impacts on aerosol variations in the Po Valley (e.g. Gilardoni et al., 2014). We consider that additional innovative information on fog impacts on aerosol variations in the Po Valley cannot be given based on our datasets.

With one-year data, weekend effects might be explored for a better interpretation of sources.

We agree with the referee that weekend effects help better interpret sources. Weekend effects have been explored and are mentioned briefly in the manuscript, which reports that i) HOA contributions are higher during weekdays than weekends, which is characteristic of traffic emissions (l. 312) and ii) BBOA contributions are conversely higher during weekends than weekdays, consistent with residential heating emissions (l. 317).

A figure showing weekday/weekend contributions and concentrations of OA factors in each season has been added in the supplementary material (Fig. S6). Precisions have been added in the text as follows:

l. 308-310: "[HOA] relative contribution is characteristic of traffic emissions, exhibiting a peak in the morning, and higher contributions during weekdays than weekends (e.g. averages of 14 and 9%, respectively, in autumn, Fig. S6)."

l. 312-315: Regarding BBOA, "A distinct daily cycle with higher contributions during night-time than daytime is observed, in addition to higher contributions during weekends than weekdays (e.g. averages of 24 and 21%, respectively, in spring, Fig. S6), consistent with residential heating emissions."

Line 394, "and previous observations (Putaud et al., 2013)" is not one of the reasons.
Modified accordingly:
"Higher levels were expected during cold months due to enhanced biomass burning emissions and lower boundary layer heights (BLH), as previously observed at the study site (Putaud et al., 2013)."

Line 449, typo, "C2H30+".
Modified accordingly, "$C_2H_3O^+$"

**References**

Gilardoni, S., Massoli, P., Giulianelli, L., Rinaldi, M., Paglione, M., Pollini, F., Lanconelli, C., Poluzzi, V., Carbone, S., Hillamo, R., Russell, L. M., Facchini, M. C. and Fuzzi, S.: Fog scavenging of organic and inorganic aerosol in the Po Valley, Atmospheric Chem. Phys., 14(13), 6967–6981, doi:10.5194/acp-14-6967-2014, 2014.

Putaud, J.-P., Adam, M., Belis, C. A., Bergamaschi, P., Cancellinha, J., Cavalli, F., Cescatti, A., Daou, D., Dell'Acqua, A., Douglas, K., Duerr, M., Goded, I., Grassi, F., Gruening, C., Hjorth, J., Jensen, N. R., Lagler, F., Manca, G., Martins Dos Santos, S., Passarella, R., Pedroni, V., Rocha e Abreu, P., Roux, D., Scheeren, B., and Schembari, C.: JRC-Ispra Atmosphere-Biosphere-Climate Integrated monitoring Station (ABC-IS): 2011 report, JRC Technical Reports, Joint Research Centre, Ispra (Italy), available at: http://publications.jrc.ec.europa.eu/repository/bitstream/111111111/28242/1/lb-na-25753-en-n.pdf (last access: 28 March 2014), 2013.

---

## Author Comment (AC2) · 29 Jul 2016

Dear Referee #2,

Please find below your comments in blue and the authors' responses in black.

This manuscript reports results obtained with an Aerodyne Aerosol Chemical Speciation Monitor (ACSM) during a long-term measurement period (1 year) in the Po Valley (Italy). The authors investigated the chemical composition of non-refractory submicron PM (NR-PM1) with a time-resolution of 30 minutes and identified of the main components of the organic fraction. In addition, the parallel multiple off-line analyses were carried out to assess the performance of the ACSM in the determination of PM chemical species regulated by Air Quality Directives. This work is meaningful, and the workload is very large. The observed results are representative because they have been measured for almost one year. It is completely within the scope of Atmospheric Chemistry and Physics. However, the results presented here are a little superficial, and a more deeply analyses on these observation results is necessary. Therefore, I recommend its publication after a major revision.

Specific Comments
Abstract
Although the abstract has summarized the main contents of the text, it is still lack of some points to attract readers. I think the authors have done a lot of works, but they do not have an in-depth analysis on their results. For example, the author did not find the BBOA in summer. While, they only showed the average results in this part and main text. Then some interesting information was averaged and masked. Therefore, I believe other more meaningful results could be found after a more in-depth analysis.

After the comments of referee #2 on 16/09/2015, the abstract has been fully reworded to better highlight innovative results and therefore attract readers. The absence of BBOA in summer is mentioned in the main text (l. 282-283), and further discussed in the supplementary material (l. 68-88; l. 103-117). We believe it has been the subject of an in-depth analysis. In this manuscript, we chose to present results which are representative of a given time-period (e.g. daily cycles, seasonal averages, etc.), in order to recommend PM abatement strategies. Specific pollution events or phenomena are therefore not presented here by choice.

Introduction
Please add a simple introduction on the previous ACSM and AMS research results in this area. Then discuss the innovation of your research is more appropriate.

A simple introduction on the previous ACSM and AMS research results in this area is added accordingly: l.73-76: "In addition, studies based on aerosol mass spectrometer measurements have been conducted in the Po valley, with the aim of characterizing specific phenomena (e.g. fog events, cooking aerosols) or seasons (Dall'Osto et al., 2015; Decesari et al., 2014; Gilardoni et al., 2014; Saarikoski et al., 2012)".

About the article structure.
Section 3 (results) contains : (1) discussion on consistency of ACSM measurements; (2) discussion on organic apportionment quality control. Meanwhile, some content looks like a introduction of analysis method. I think these contents are more like a discussion part not results. On the contrary, the author showed a lot of experimental results in section 4 (discussion). Therefore, I suggest that the author reconsider the article structure carefully and some discussion content could be put in the supplementary material.

The titles of Sect. 3, 3.1, 3.2 and 4 have been reconsidered. The new titles are: Sect. 3 "Quality assurance / quality control", Sect. 3.1. "Quality assurance / quality control of ACSM measurements", Sect. 3.2. "Quality assurance / quality control of organic source apportionment", and Sect. 4 "Results and discussion".

About the language: Some sentences are not easy to understand. While high quality language expression is very important for a high quality article. Therefore, please improve the language after this modification.

We did go through the manuscript and corrected language issues where we found them. In addition the publication shall be proof-read by the editorial office of ACP.

At the beginning of section 4.3, the variations of NR-PM1 chemical composition and OA factors' contributions as a function of total NR-PM1 mass are examined. However, these analysis is not deep enough. According to the results mentioned above, the contribution of every NR-PM1 and OA species to the total is different in four seasons. It means that the pollution characteristics in four seasons may be different. Therefore, I strongly recommend the author to discuss the variation of every species with the change of total NR-PM1 mass for different seasons. Then some common features for four seasons and some unique characteristics for one season could be found. These results are more interesting than that provided now.

The variation of every species with the change of total NR-PM$_1$ mass has been investigated for different seasons accordingly (Fig. S9, Sect. S4). The following text has been added in the manuscript:

l. 496-499: "In order to investigate the characteristics of fine aerosol pollution events, the variations of NR-PM1 chemical composition and OA factors' contributions as a function of total NR-PM1 mass are examined. This investigation is made on the annual (Fig. 8, discussion below) and seasonal scales (Fig. S9, discussion in Sect. S4)."

Supplementary material: Sect. S4 has been added as follows:

l. 131-142: "The seasonal variations of NR-PM$_1$ chemical composition and OA factors' contributions as a function of total NR-PM$_1$ mass is shown in Fig. S9. During spring and autumn, the characteristics of submicron aerosol pollution events are similar to what is described on the annual scale (Sect. 4.3). Nitrate and BBOA are the main responsible for NR-PM$_1$ concentrations increase. Specific patterns are observed during summer, with i) fairly stable contributions of OA sources irrespective of NR-PM$_1$ levels, and ii) noticeable proportions of sulfate (~20%). Sulfate is nevertheless not responsible for the increase of NR-PM$_1$ concentrations. Note that the intensity and frequency of NR-PM$_1$ pollution events are lower during summer compared to other seasons. Finally, the importance of BBOA in submicron aerosol pollution events is highlighted during winter ([BBOA]~40% of [OA], when [NR-PM$_1$]> 30 µg/m$^3$). An increase of HOA is for the first time observed during this season, which indicates that primary sources largely contribute to high NR-PM$_1$ concentrations during winter."

Figure 3:
(1) "Left, bottom, top right, bottom right" is not clear enough. Figure 3a, b, c, d is better. I suggest the author do the modification in this and other figures.

Figure 3 has been modified accordingly (Fig. 3 a, b, c, d).
Figure 5 has been modified similarly.

(2) This figure looks not every clear.
For example:
The following figure is a part of the relative chemical composition of all NR-PM1 species. I found that the contribution of organic in spring (March-April-May) is lower than 50% (red line) in most of time, and frequently lower than 40%. However, the average contribution reached to 52% in the main text. It looks very strange.

In addition, according to the relative chemical composition figure, it seems that the contribution of organic was 0 some time. However, the mass concentration of organic looks always higher than 0. The same phenomenon also appeared in Figure 5. Please see the following figure: It can be found that the contribution of HOA is almost always higher than 10% (red line), even frequently higher than 30%. However, the average contribution was only 11%.

Referee#2 is thanked for pointing out this issue. There were indeed problems with Fig. 3 and 5 exhibiting the relative contributions of chemical species and OA factors with 30 min time-resolution. These figures have been updated. We confirm that the average contributions mentioned in the text, on the seasonal and annual scales, were correct.

Other comments:
Abstract.
Line 31-33. A detail comparison with other study results is not necessary in abstract. I don't recommend to cite the detailed data results.
Detailed data results - i.e. "(USA, 14.2 µg/m$^3$)" and "(Japan, 12-15 µg/m$^3$)" - have been removed accordingly.

Introduction.
Line 50 Please provide the annual PM2.5 mean concentration value. That could make this sentence more convincing.
PM2.5 mean concentration values are provided in this sentence for an urban (l. 52) and a regional background site (l. 53).

Line 69 "2008/50/EC" is a reference or other standard ? It is not easy to understand.
"2008/50/EC" refers to the European Directive mentioned in this sentence. For clarity, this has been replaced by a reference in the text: "(EU, 2008)" (please see references at the end of our responses).

Section 2.1 I suggest the author add a sampling site map in main text or supplement material.
A sampling site map has been added in the supplement material accordingly (Fig. S1, reference in the main text l. 111).

Line 205 Please check the format.
Format modified accordingly.

Line 219 What is the reason for the poor correlation and results difference of chloride ?
Chloride quantification suffers from high uncertainties, both from ion chromatography analyses (l. 248) and from the ACSM (Crenn et al., 2015).

Section 3.2.3 Although the author explained the reasons for HOA was not well correlated with BC, CO and NOx, I still feel confused. What are the main sources of BC, CO, and NOx in the study site ? It looks BBOA has a very big impact on them. In addition, chloride is also an important tracer for BBOA, did you have an analysis of their relationship ?
The main sources of BC, CO and NOx at the study site are fossil fuel and biomass burning. BBOA indeed has a very big impact on them. For example, during winter, Gilardoni et al. (2011) estimated that fossil fuel and biomass burning sources contribute to 51 and 49%, respectively, of Elemental Carbon concentrations at the study site during winter. Gaeggeler et al. (2008) also report high correlations between CO and BC stemming from biomass burning in the alpine valleys.
Chloride in the form of potassium chloride is indeed an important tracer for BBOA. The ACSM analyses non-refractory (NR-) chloride i.e. mostly ammonium chloride (Huang et al., 2010). We thus do not expect to have correlations between BBOA and NR-Chloride here because biomass burning related chloride would be mostly in the form of KCl. As mentioned before, high uncertainties are associated with chloride quantification from ion chromatography (IC) analyses, preventing us from making comparisons between BBOA and IC-Chloride.

Line 376-378 Pleases add the values of the highest NR-PM1 levels reported at rural and urban …….
The highest NR-PM$_1$ level reported i) in Europe (Crippa et al., 2014) is 16.4 µg/m$^3$ in Barcelona (Spain), ii) in the world (Jimenez et al., 2009; Zhang et al., 2007, 2011) is 63 µg/m$^3$ in Beijing (China). We found

misleading to mention only the maximum NR-PM$_1$ levels reported in the previous studies, since it is stated that: "the annual averaged NR-PM$_1$ mass reported here (14.2 μg/m$^3$) ranges amongst the highest NR-PM$_1$ levels reported at rural and urban downwind sites in Europe (Crippa et al., 2014) and worldwide (Jimenez et al., 2009; Zhang et al., 2007, 2011)". We have nonetheless reported that our site has the 7th highest NR-PM1 levels out of 41 sites compared in the aforementioned studies. We also have mentioned that these previous studies are based on typically one month of measurements in different seasons.

Line 385-386 Please add "(Fig. 3)" to the back of this sentence.
"(Fig. 3)" is added to the back of this sentence accordingly.

Line 393 Did you measured the BLH in different seasons ? A comparison on measurement results is more useful.
BLH were not measured in different seasons during the campaign. The fact that BLH are lower during cold months in the Po Valley is well documented in the literature (e.g. Vecchi et al., 2004).

Figure 1 and 2. These figures are very difficult to see clearly due to the too small words. It is necessary to enlarge the font size in the two figures.
The font size has been enlarged in Fig. 1 and Fig. 2 accordingly.

Figure 6. It is not very clear and the font is too small. I suggest that all the figures are divided into two lines. Then the figures will not too narrow.
Figure 6 has been divided into two lines accordingly.

**References**

[revised manuscript text omitted]

---

## Author Comment (AC3) · 29 Jul 2016

Dear Referee #3,

Thank you for your comments on our manuscript. Please find below your comments in blue and the authors' responses in black.

This paper presents an analysis of ACSM data from the Po Valley, Italy, using ME-2 factorisation. While this is becoming an increasingly common form of measurement and analysis, this paper remains relevant because it is the first such analysis of one of the most polluted rural sites in Europe and may have implications for regional pollution in this area. The results aren't particularly surprising and the technical developments are incremental at best, however the results and quality assurance data are presented in a very comprehensive manner and the results analysed in the context of air quality control policy, so it makes an overall contribution to the science in these regards. Overall, this paper is well-written and I would recommend publication after the authors consider the following minor comments.

General comments:
The paper currently lacks a comparison with other AMS factorisations done at this site, specifically Decesari et al. (2014, doi:10.5194/acp-14-12109-2014) and Dall'Osto et al. (2015, doi:10.1021/acs.est.5b02922). This strikes me as a major omission.
The studies of Decesari et al. (2014) and Dall'Osto et al. (2015) are reported in the revised version of the manuscript accordingly. However, please note that both studies were conducted at a different south-eastern site of the Po Valley (San Pietro Capofiume, ca. 300km distant) and for short-time windows representing single seasons, hence limiting comparisons with our study. The following paragraphs have been added or modified:
l.73-76 (paragraph added in agreement with referee#2 suggestions as well):
"In addition, studies based on aerosol mass spectrometer measurements have been conducted in the Po valley, with the aim of characterizing specific phenomena (e.g. fog events, cooking aerosols) or seasons (Dall'Osto et al., 2015; Decesari et al., 2014; Gilardoni et al., 2014; Saarikoski et al., 2012)".
l.283-286: "Note that COA could not be evidenced, likely due to the type of site studied (regional background) and the lower sensitivity, time- and mass-to-charge-resolution of the ACSM compared to classical AMS instruments (further discussed in Sect. S2; see also Dall'Osto et al., 2015 on this subject)."
l. 406-409: "A distinct peak is however observed around noon, probably caused by enhanced photochemical production of secondary organic compounds, and increased BLH favouring downward mixing of advected pollution, especially during summer (Fig. 4; Decesari et al., 2014)."
l. 416-420: "This observation could be due to i) local production of sulfate with increased photochemical production around noon at the study site and/or ii) diurnal changes of the atmospheric stratification in the Po Valley as described by Saarikoski et al. (2012) and Decesari et al. (2014), enhancing aged particle contribution during the middle of the day and the afternoon."
Supplementary Material, l.66-67:
"Further discussion on the presence (or absence) of cooking factors at rural sites of the Po Valley can be found in Dall'Osto et al. (2015)."

I am struggling to see what the mass spectral marker analysis in section 4.2 contributes to the conclusions of the paper. The analysis exhibits behaviours broadly similar with the results from the factorisation and while speculative conclusions are offered for the behaviours, these are largely inconclusive. The section could do with being shorter and more focused on the analyses that result in new scientific insight.
We wanted to present a mass spectral marker analysis in this study since it is the first time mass spectra measurements of organic aerosols are reported in the upper Po Valley. The main information presented in this section - which differs from the factorisation analysis - are:
  i)     acid species dominate the OA composition with respect to non-acid oxygenates;

ii)     most acid and non-acid oxygenates have been formed before reaching the study site, i.e. have been imported from other regions;

iii)    a first estimation of the oxygen-to-carbon (O/C), OM-to-OC (OM/OC), hydrogen-to-carbon (H/C) ratios and the carbon oxidation state (OSc) is made.

We nevertheless agree with the referee that this section do not contribute significantly to the main conclusions of the paper, and its results are consequently only briefly mentioned in the abstract or the conclusion sections. Following the referee suggestions, this section 4.2 has been shortened as follows:

Former l.459-460, removed: "favoured by low temperatures as previously found for semi-volatile OOA";

Former l.462-463, removed: ", although the amplitude of their daily cycles is less pronounced than that of acid species (~1 and 3%, respectively)";

Former l.466-469, figures removed: "(~2%)", "(~1%)" and "medians of 13-16% and 7-8% depending on the time of the day, respectively";

Former l. 481-490, paragraph moved in the supplementary material as a Section S3, and replaced by:

"Uncertainties associated with these estimates - in particular based on ACSM measurements - are discussed in Sect. S3. Comparisons with studies using (HR-ToF-) AMS instruments will not be reported and only variations within this dataset will be discussed (see Sect. S3)."

Specific comments:
Line 328: Given that m/z=60 results from a primary emissions from biomass burning, it is possible that its presence in OOA is more likely due to factor mixing than SOA production. Such an issue is very possible, given the variations within BBOA and the mass spectral resemblance of primary HULIS to LV-OOA (e.g. http://www.atmos-chemphys.net/15/2429/2015/).
Heringa et al. (2011) and Cubison et al. (2011) report that m/z=60 can be found in secondary biomass burning OA. Similarly, HULIS can be associated with secondary BBOA generation, as e.g. mentioned by Graber et al. (2006):
"Different suggested mechanisms for HULIS generation during biomass burning include: (i) soil-derived humic matter lofted into the air as a result of combustion; (ii) HULIS generation via chemical transformations during combustion and thermal breakdown of plant lignins and cellulose; and (iii) recombination and condensation reactions between volatile, low molecular weight combustion products (Mayol-Bracero et al., 2002)."
We also agree with the referee that the mass spectral resemblance of primary HULIS to LV-OOA might explain the previous observation. The following sentence has been added in the text accordingly:
l. 328-330: "Note that the mass spectral resemblance of primary humic-like substances to LV-OOA might also partly explain this observation (e.g. Young et al., 2015), i.e. that a small fraction of primary OA is found in this factor."

Line 349: While a good correlation between the BBOA factor with 60 and 73 is worth reporting, this only indicates that the factor follows these markers; to take this as a sign of accuracy, one must assume that these markers are accurate reflections of actual BBOA, which may or may not be the case.
We indeed assume that 60 and 73 are accurate markers of biomass burning here, referring to Lee et al. (2010) and references therein (l. 312).

Line 362: None of these tests do not exclude the possibility that there is a degree of rotational freedom between factors. Such freedoms can change magnitudes of signals without significantly changing their time series. It's also possible that exchanges between factors can be via a third factor (e.g. OOA).
We agree that a degree of rotational freedom between factors cannot be excluded in PMF analyses. This specific reason has been mentioned in the manuscript as follows:
"Although uncertainties associated with the accurate apportionment of HOA and BBOA cannot be excluded (e.g. due to rotational ambiguity), (…)".

We however believe that the a-value sensitivity test (third test mentioned) partly assesses the degree of rotational freedom influencing the separation of factors. In fact, decreasing the a-value (i.e. the factor profile constraint) leads to higher rotational freedom. Figure S4 shows that a-values variations have little influence on factor contributions, which means that the reported results appear to be rather robust.

Line 397: Other reasons for a seasonal high of sulphate are plausible, such as changes in source regions due to seasonal changes in the prevailing wind direction, or changes in the amount of rainout.
We agree that both reasons mentioned by the referee are plausible. The sentence line 397 has been tempered as follows:
"Expected seasonal variations of the chemical composition of NR-PM$_1$ are observed, with (…) higher sulfate contributions during summer, which can e.g. be associated with enhanced photochemical production (Seinfeld and Pandis, 2006) and lower amount of rainout (Fig. S5) (…)."

Line 407: The statement that the midday peak is due to in situ photochemistry is at odds with the discussion towards the end of the paragraph, where this is rightfully treated with scepticism. This could be tested by comparing SO4 with SO2 and looking for a diurnal pattern in the fraction of oxidised sulphur as SO4. However, I would expect it more likely that the peak is due to the increased PBL height during the day favouring downward mixing of advected pollution. As pointed out, the timescale of formation of SO4 is too long to expect a pattern like this to result from chemistry.
This sentence has been revised accordingly as follows:
"A distinct peak is however observed around noon, probably caused by enhanced photochemical production of secondary organic compounds, and increased BLH favouring downward mixing of advected pollution, especially during summer (Fig. 4; Decesari et al., 2014)."

Technical comments:
While conventional, the definitions of the seasons used should be given in the main text rather than just a figure caption.
The following sentence has been added in the main text (l. 144) accordingly:
"Seasons are defined as spring (MAM), summer (JJA), autumn (SON) and winter (DJF)."

Line 178: Please specify the 'classical program' used for PMF.
The 'classical programs' used for PMF have been specified as follows:
"Contrary to classical programs used to resolve PMF (e.g. PMF2, PMF3), ME-2 allows any element of the F and G matrices to be constrained with a certain degree of freedom."

SI line 20: Please use scientific rather than engineering notation.
Scientific rather than engineering notation has been used accordingly.

SI line 102: Correct "Error! Reference source not found."
Corrected:
"Correlations with independent measurements are further discussed in Sect. 3.2 of the manuscript."

Figure S5: Do the curved lines on these plots represent actual data or a nonlinear interpolation between points? If it is the latter, the algorithm should be specified and justified.
The curved lines represented a nonlinear interpolation between points in the previous version of the manuscript. Curved lines are removed in the updated figure.

**References**

Cubison, M. J., Ortega, A. M., Hayes, P. L., Farmer, D. K., Day, D., Lechner, M. J., Brune, W. H., Apel, E., Diskin, G. S., Fisher, J. A., Fuelberg, H. E., Hecobian, A., Knapp, D. J., Mikoviny, T., Riemer, D., Sachse, G. W., Sessions, W., Weber, R. J., Weinheimer, A. J., Wisthaler, A., and Jimenez, J. L.: Effects of aging on organic aerosol from open biomass burning smoke in aircraft and laboratory studies, Atmos. Chem. Phys., 11(23), 12049–12064, doi:10.5194/acp-11-12049-2011, 2011.

Dall'Osto, M., Paglione, M., Decesari, S., Facchini, M. C., O'Dowd, C., Plass-Duellmer, C., and Harrison, R. M.: On the Origin of AMS "Cooking Organic Aerosol" at a Rural Site, Environ. Sci. Technol., 49(24), 13964–13972, doi:10.1021/acs.est.5b02922, 2015.

Decesari, S., Allan, J., Plass-Duelmer, C., Williams, B. J., Paglione, M., Facchini, M. C., O'Dowd, C., Harrison, R. M., Gietl, J. K., Coe, H., Giulianelli, L., Gobbi, G. P., Lanconelli, C., Carbone, C., Worsnop, D., Lambe, A. T., Ahern, A. T., Moretti, F., Tagliavini, E., Elste, T., Gilge, S., Zhang, Y., and Dall'Osto, M.: Measurements of the aerosol chemical composition and mixing state in the Po Valley using multiple spectroscopic techniques, Atmos. Chem. Phys., 14, 12109–12132, doi:10.5194/acp-14-12109-2014, 2014.

Gilardoni, S., Massoli, P., Giulianelli, L., Rinaldi, M., Paglione, M., Pollini, F., Lanconelli, C., Poluzzi, V., Carbone, S., Hillamo, R., Russell, L. M., Facchini, M. C., and Fuzzi, S.: Fog scavenging of organic and inorganic aerosol in the Po Valley, Atmos. Chem. Phys., 14(13), 6967–6981, doi:10.5194/acp-14-6967-2014, 2014.

Graber, E. R. and Rudich, Y.: Atmospheric HULIS: How humic-like are they? A comprehensive and critical review, Atmos Chem Phys, 6(3), 729–753, doi:10.5194/acp-6-729-2006, 2006.

Heringa, M. F., DeCarlo, P. F., Chirico, R., Tritscher, T., Dommen, J., Weingartner, E., Richter, R., Wehrle, G., Prévôt, A. S. H., and Baltensperger, U.: Investigations of primary and secondary particulate matter of different wood combustion appliances with a high-resolution time-of-flight aerosol mass spectrometer, Atmos. Chem. Phys., 11(12), 5945–5957, doi:10.5194/acp-11-5945-2011, 2011.

Lee, T., Sullivan, A. P., Mack, L., Jimenez, J. L., Kreidenweis, S. M., Onasch, T. B., Worsnop, D. R., Malm, W., Wold, C. E., Hao, W. M., and Collett, J. L.: Chemical smoke marker emissions during flaming and smoldering phases of laboratory open burning of wildland fuels, Aerosol Sci. Tech., 44(9), i–v, doi:10.1080/02786826.2010.499884, 2010.

Mayol-Bracero, O. L., Guyon, P., Graham, B., Roberts, G., Andreae, M. O., Decesari, S., Facchini, M. C., Fuzzi, S., and Artaxo, P.: Water-soluble organic compounds in biomass burning aerosols over Amazonia – 2. Apportionment of the chemical composition and importance of the polyacidic fraction, J. Geophys. Res.-Atmos., 107, D20 8091, doi:10.1029/2001JD000522, 2002.

Saarikoski, S., Carbone, S., Decesari, S., Giulianelli, L., Angelini, F., Canagaratna, M., Ng, N. L., Trimborn, A., Facchini, M. C., Fuzzi, S., Hillamo, R., and Worsnop, D.: Chemical characterization of springtime submicrometer aerosol in Po Valley, Italy, Atmos. Chem. Phys., 12(18), 8401–8421, doi:10.5194/acp-12-8401-2012, 2012.

Seinfeld, J. H. and Pandis, S. N.: Atmospheric Chemistry and Physics: from Air Pollution to Climate Change, Wiley, New York, USA, 2006.

Young, D. E., Allan, J. D., Williams, P. I., Green, D. C., Harrison, R. M., Yin, J., Flynn, M. J., Gallagher, M. W. and Coe, H.: Investigating a two-component model of solid fuel organic aerosol in London: processes, PM1 contributions, and seasonality, Atmos Chem Phys, 15(5), 2429–2443, doi:10.5194/acp-15-2429-2015, 2015.

---

## Author Response (AR2)

Comments to the Author:
The authors have reasonably addressed the comments of the three anonymous referees.
However, I have a number of (mostly minor) comments that need to be addressed before this
manuscript can be published in ACP.
Response to the Co-Editor:
The authors have addressed the comments of the Co-Editor, as detailed below.

For the main text:
Line 80: Replace "their source" by "its sources".
Replaced accordingly.
Line 99: Replace "30 min" by "30-min".
Replaced accordingly.
Line 124: Replace "30 minutes" by "30-min".
Replaced accordingly.
Line 129: Replace "70eV" by "70 eV".
Replaced accordingly.
Line 154: Replace "are sampled" by "was sampled".
Replaced accordingly.
Line 175: Replace "Briefly PMF" by "Briefly, PMF".
Replaced accordingly.
Line 177: Replace "to classical" by "to the classical".
Replaced accordingly.
Line 212: Replace "fractions>40%" by "fractions >40%".
Replaced accordingly.
Line 220: Replace "ACSM also" by "ACSMs also".
Replaced accordingly.
Line 382: Replace "guidelines given" by "guideline given".
Replaced accordingly.
Lines 404, 412, 415, 455, 456, and 466: Replace "night time" by "night-time".
Replaced accordingly.
Line 442: Replace "variations show" by "variations shows".
Replaced accordingly.
Line 459: Replace "the decreasing of" by "the increasing of".
Replaced accordingly.
Line 465: Replace "characteristics of" by "characteristic of".
Replaced accordingly.
Line 1052: Replace "Slopes of mass vs volume are in g/cm3 and" by "Slopes are in g/cm3 for mass vs
volume and".
Replaced accordingly.
Line 1084: Replace "averages in" by "averages as a".
Replaced accordingly.
Line 1085: Replace "of events" by "of pollution events".
Replaced accordingly.

For the Supplementary Material:
Line 22: Replace "represents the" by "represent the".
Replaced accordingly.
Line 32: Replace "(Crenn et al., 2015)" by "Crenn et al. (2015)".
Replaced accordingly.
Line 104: Replace "for SA analysis" by "for source apportionment (SA) analysis".
Replaced accordingly.
Line 150, Table S2: The data in this table are hard to read; I suggest increasing the font size.

The font size of Table S2 has been increased accordingly.
Line 185: "Kroll et al. (2011)" is missing in the Reference list.
"Kroll et al. (2011)" has been added in the reference list accordingly.
Line 187: Replace "in function" by "as a function".
Replaced accordingly.
Line 188: Replace "2 events" by "2 pollution events".
Replaced accordingly.